# A cytoplasmic form of EHMT1$^N$ methylates viral proteins to enable inclusion body maturation and efficient viral replication

**Kriti Kestur Biligiri**[1,2], **Nishi Raj Sharma**[3], **Abhishek Mohanty**[4], **Debi Prasad Sarkar**[5], **Praveen Kumar Vemula**[4], **Shravanti Rampalli**[1,2]*

1 Council of Scientific and Industrial Research (CSIR)–Institute of Genomics and Integrative Biology (IGIB), New Delhi, India, 2 Academy of Scientific and Innovative Research (AcSIR), Ghaziabad; India, 3 Department of Education and Research, AERF, Artemis Hospitals, Gurugram, India, 4 Institute for Stem Cell Science and Regenerative Medicine (inStem), GKVK Campus, Bangalore, India, 5 Department of Biological Sciences and Engineering, Indian Institute of Technology, Gandhinagar, Palaj, Gujarat, India

* shravanti@igib.in, shravanti@igib.res.in

**Data Availability Statement:** The summary data underlying the figures containing Western blots, graphs and tables are provided as S1 Data. Source

## Abstract

Protein lysine methyltransferases (PKMTs) methylate histone and non-histone proteins to regulate biological outcomes such as development and disease including viral infection. While PKMTs have been extensively studied for modulating the antiviral responses via host gene regulation, their role in methylation of proteins encoded by viruses and its impact on host–pathogen interactions remain poorly understood. In this study, we discovered distinct nucleo-cytoplasmic form of euchromatic histone methyltransferase 1 (EHMT1$^{N/C}$), a PKMT, that phase separates into viral inclusion bodies (IBs) upon cytoplasmic RNA-virus infection (Sendai Virus). EHMT1$^{N/C}$ interacts with cytoplasmic EHMT2 and methylates SeV-Nucleoprotein upon infection. Elevated nucleoprotein methylation during infection correlated with coalescence of small IBs into large mature platforms for efficient replication. Inhibition of EHMT activity by pharmacological inhibitors or genetic depletion of EHMT1$^{N/C}$ reduced the size of IBs with a concomitant reduction in replication. Additionally, we also found that EHMT1 condensation is not restricted to SeV alone but was also seen upon pathogenic RNA viral infections caused by Chandipura and Dengue virus. Collectively, our work elucidates a new mechanism by which cytoplasmic EHMT1 acts as proviral host factor to regulate host–pathogen interaction.

## Introduction

Euchromatic histone N-lysine methyltransferase 1 (EHMT1) is an enzyme that belongs to the SET domain family of methyltransferases [1], known for its dual activity of methylating lysine residues [1] as well as binding methylated lysine residues [2]. EHMT1, along with its homolog, EHMT2, forms heteromeric complexes [1,3] to methylate the ninth lysine residue of Histone 3 (H3K9), thereby transcriptionally repressing regions of the chromatin and signalling the formation of heterochromatin [1,4]. The SET domain of EHMTs catalyse the transfer of methyl

data for all the graphs and tables are provided as numerical values in separate excel sheets. Each excel sheet corresponds to a graph or table, the sheets are labelled accordingly. This file is labelled S1 Data. All the raw western blots are provided in their original unedited form in a PDF document. This file is labelled S1_raw_images. The raw Confocal imaging data is uploaded Bioimage Archive (https://www.ebi.ac.uk/biostudies/bioimages/studies/S-BIAD1362) with accession number S-BIAD1362.

**Funding:** This project was supported by funds from DBT/Wellcome Trust India Alliance Intermediate Fellowship (Grant number # 500220-Z-11-Z) to SR, funds from CSIR-IGIB (OLP 1153 and OLP 2302) to SR and inStem Core funds to PKV. K.K.B is supported by CSIR-JRF/SRF fellowship. NRS received support from the Ramalingaswami Re-entry Fellowship from Department of Biotechnology (DBT), Ministry of Science and Technology, Government of India (BT/RLF/Re-entry/40/2018). The funders had no role in study design, data collection and analysis, decision to publish, or preparation of the manuscript.

**Competing interests:** The authors have declared that no competing interests exist.

**Abbreviations:** DI, defective interfering; EHMT1, euchromatic histone methyltransferase 1; HDF, human dermal fibroblast; IB, inclusion body; MEF, mouse embryonic fibroblast; PIC, protease inhibitor cocktail; PKMT, protein lysine methyltransferase; PTM, posttranslational modification; SeV, Sendai virus; WT, wild-type.

groups to lysine residues [1], and the ankyrin (Ank) domain recognises and binds the methyl groups [2]. In addition to histones, EHMTs methylate non-histone substrates to modulate a broad range of biological processes [5–9]. Additionally, a high-throughput study performed to determine the enzymatic substrates of EHMT1 and EHMT2 resulted in the identification of overlapping and non-overlapping substrates [6]. Among these, several proteins were extranuclear, including several mitochondrial, ER, and cell membrane-specific proteins [6]. While cytoplasmic isoform of EHMT2 is known [10–12], methylation of this vast array of EHMT1 specific extranuclear substrates could not be explained as it was thought to be restricted to the nucleus [8]. The histone methyltransferase activity of EHMTs has been extensively characterised in modulating the host antiviral response. For example, studies in embryonic stem cells identified that EHMT2 is required for exogenous retroviral silencing and MERVL ERVs [13,14]. EHMT2-mediated H3K9me2 activity represses proinflammatory gene (IFN and IFN signalling genes) expression and upon viral infection this repression is relieved to resist the viral infection [15]. The histone lysine methyltransferase activity of EHMT1 has also been characterised in modulating the host antiviral responses [15,16]. For instance, EHMT1 maintains latency of retroviruses like HIV-1 by transcriptionally silencing the proviral elements integrated into the host genome [17]. EHMT1 also regulates the expression of various inflammatory cytokines like IFNβ to maintain homeostasis under normal conditions [15,16]. During infection by cytoplasmic RNA viruses like Influenza, Sendai, and VSV, IFNβ expression is increased by releasing the repression laid by EHMT1 to inhibit viral replication [16]. Overall, EHMTs are known to play an active epigenetic role during RNA viral infections by regulating the host antiviral response. However, direct involvement of EHMTs in modulating RNA viral replication via its non-histone protein methylation has not been reported till date.

While performing cellular reprogramming experiments using Sendai virus (SeV) as a vector to deliver OSKM factors, we observed cytoplasmic condensation of EHMT1. Since EHMT1 is a nuclear protein, its unanticipated cytoplasmic localization led to further investigations, during which, we identified novel nucleo-cytoplasmic forms of EHMT1 (EHMT1$^{N/C}$), EHMT1$^{B1-3}$, and EHMT1$^{V09}$. We found that EHMT1$^{N/C}$ acts as proviral host factor which was recruited to inclusion bodies (IBs) of SeV. At mechanistic level, we discovered that cytoplasmic EHMTs (EHMT1$^{N/C}$ and EHMT2) interact with and methylate the nucleoprotein of SeV upon infection. Methyltransferase activity of EHMTs facilitated the formation of large replication platforms, thereby enhancing viral replication and propagation. Inhibition of EHMT's enzymatic activity by small molecule inhibitors or depleting the levels of EHMT1$^{N/C}$ by CRISPR/Cas9 system revealed loss of larger IBs in infected cells indicating impairment in growth or coalescence. Accordingly, we also noticed reduced replication of Sendai viral genomic RNA in infected cells that were treated with EHMT inhibitors or gene edited. Overall, in this study we demonstrate for the first time, novel nucleocytoplasmic forms of EHMT1 that enzymatically promotes RNA viral replication via methylation of the nucleoprotein.

# Results

## Sendai virus induces cytoplasmic condensation of EHMT1

Ectopic expression of Yamanaka factors (OSKM) in somatic cells rewires the global epigenetic landscape towards pluripotent state [18–20]. Such dramatic alterations are achieved by combinatorial efforts of several chromatin modifiers including EHMT1 [21,22]. While the canonical role of EHMT1 has been studied during reprogramming, its non-canonical role remains poorly understood. Our study began with the intent to identify the non-canonical substrates of EHMT1 during the early phase of iPSC generation. Towards this, we adopted integration free [Cytotune-OSKM (Sendai viral vector) and episomal OSKM (plasmid based)] transgene

delivery systems that are widely used in the field. We examined the pattern of expression of EHMT1 in (1) fibroblasts; (2) heterogenous cultures of OSKM transduced fibroblasts; and in (3) iPSCs, by performing immunolabelling for EHMT1. Fibroblasts and iPSCs, demonstrated nuclear localization of EHMT1 (S1A Fig), which was consistent with its previously known sub-cellular localisation [8]. Interestingly, in the population of fibroblasts transduced with OSKM factors via SeV, we observed EHMT1 condensates of heterogenous size and shape distributed throughout the cytoplasm in addition to its expression in the nucleus (S1A Fig). This was observed around Day 5, in epithelial-like colonies amidst the mesenchymal fibroblasts. Since reprogramming is a long process that usually takes about 3 weeks, we profiled the formation of these condensates at various time points. At 12 h post transduction of the reprogramming factors, these condensates formed as tiny speckles (S1B Fig), distributed in the cytoplasm, which then progressed to form larger condensates, which peaked in size around Day 3 (S1B Fig), post which, its size reduced and by Day 10, they appeared to have been mostly resolved (S1B Fig). This experiment indicated that condensates formed by EHMT1 appeared around 12 h and persisted in the cytoplasm till about Day 10 (S1B Fig), although their size was greatly reduced by the 10th day. Co-immunolabelling the heterogenous cultures with EHMT1 and Oct4 confirmed cytoplasmic condensation of EHMT1 in OSKM transduced cells (S1C Fig). However, surprisingly, such cytoplasmic condensates were completely absent in reprogramming cultures ectopically expressing OSKM via episomal plasmids (S1D Fig), indicating that the cytoplasmic localization of EHMT1 was not in response to OSKM expression. Since the localization of EHMT1 differed among the 2 modes of reprogramming, we speculated that its cytoplasmic condensation was not induced by reprogramming but was in response to the vector, Sendai virus. To test this, we infected fibroblasts with Cytotune EmGFP reporter SeV or wild-type (WT) SeV and performed immunolabelling for EHMT1. Cells infected with EmGFP SeV or WT SeV formed cytoplasmic condensates of EHMT1 of varying sizes and shapes (Fig 1, 1A and 1B), while retaining its nuclear localization, thus confirming our hypothesis. Additionally, we observed cytoplasmic EHMT1 condensation in other cell types like HEK (Figs 1C and S1E), BEAS-2B (Figs 1D and S1F) and mouse embryonic fibroblasts (MEFs) (S1G Fig) only upon SeV infection, indicating that our observations were not cell type specific. We did not detect localization of EHMT2 or Ezh2, another epigenetic modifier into viral IBs in response to viral infection (S1H and S1I Fig) indicating that this was not a common observation among all the epigenetic modifiers. Altogether, we identified that SeV infection induced cytoplasmic condensation of EHMT1, an epigenetic modifier, which was previously reported to be exclusively localized in the nucleus. Intrigued by the altered localization of EHMT1 in response to viral infection, we continued our investigation to uncover the role of EHMT1 in Sendai viral life cycle.

## EHMT1 is recruited to Sendai viral inclusion bodies

Sendai virus, a member of the paramyxoviridae family of single-strand negative-sense RNA viruses, replicates in the cytoplasm of the host [23–25]. Its genomic RNA is bound by the Nucleoprotein (N), a ssRNA binding protein, Phosphoprotein (P), a cofactor of the polymerase, and the RNA-dependent RNA Polymerase (RdRP or L) to constitute an RNP complex. Upon entering the host, the viral RNP complex of most members of the paramyxoviridae family serves as a template for transcription [26–29]. As the concentration of N, P and L proteins increase, they tend to phase separate from the aqueous cytosol, forming subcellular structures known as IBs [26–29]. By restricting the exchange of material between IBs and cytosol, these structures protect the viral components from being detected and degraded by the host immune surveillance machinery [30,31]. N and P proteins also recruit several host factors to IBs

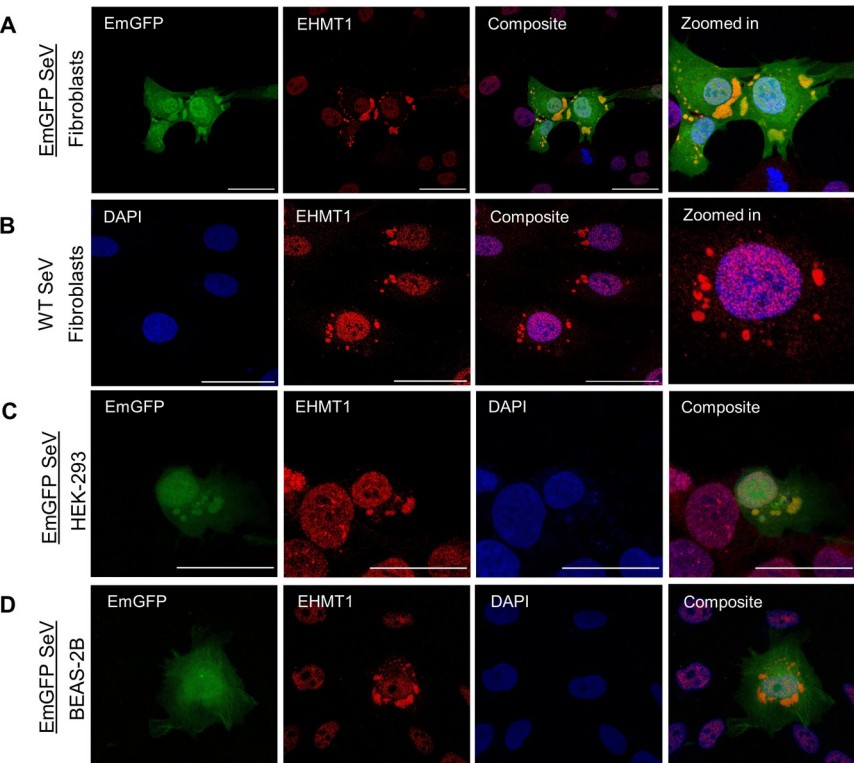

**Fig 1. SeV induces cytoplasmic condensation of EHMT1.** Confocal microscopic images of (A) HDFs infected with EmGFP SeV (green), (B) HDF infected with WT SeV immunolabelled for EHMT1 (red). (C) HEK-293 cells infected with EmGFP SeV (green), immunolabelled for EHMT1 (red). (D) BEAS-2B cells infected with EmGFP SeV (green), immunolabelled for EHMT1 (red) 48 h p.i. Composite images are with DAPI (blue) stained nuclei. Scale bar, 40 μm. Raw confocal microscopic images are deposited on BioImage Archive (Accession id: S-BIAD1362). EHMT1, euchromatic histone methyltransferase 1; HDF, human dermal fibroblast; SeV, Sendai virus; WT, wild-type.

[30–35], thus accumulating the necessary viral and host components within these subcellular structures for viral replication. While the role of viral components in IB formation has been studied, the host factors recruited to IBs and their contribution to IB formation and viral replication needs further exploration.

IBs formed by members of the paramyxoviridae family like measles virus are cytoplasmic condensates of irregular shapes and sizes. Since EHMT1 formed similar condensates in the cytoplasm in response to SeV infection, we examined if these are SeV IBs. WT SeV or EmGFP SeV infected cells were co-immunolabelled with EHMT1 and a polyclonal SeV antibody, where we found that EHMT1 colocalized with SeV (Figs 2A and S2A), indicating that SeV proteins and EHMT1 localize to the same subcellular compartment. To determine if these structures are indeed IBs, we used SeV IB-specific markers like the N and P proteins, which were cloned into an mCherry_C1 plasmid (mCh_N) and piRFP_N1 plasmid (piRFP_P), respectively. Transfection of mCh_N resulted in diffused cytoplasmic expression of N and EHMT1 (S2B and S2C Fig) but upon infection, cytoplasmic condensates of mCh_N were observed (Fig 2B), marking the IBs. ICC against EHMT1 demonstrated a colocalization of EHMT1 with SeV N in viral condensates (Fig 2B). Similarly, piRFP_P and EHMT1 when transfected independently, was diffused in the cytoplasm (S2D and S2E Fig) but condensed into SeV IBs upon infection, where EHMT1 was found to localize (Fig 2C). Thus, in SeV infected cells, EHMT1 condensed into viral IBs, marked by the N and P proteins.

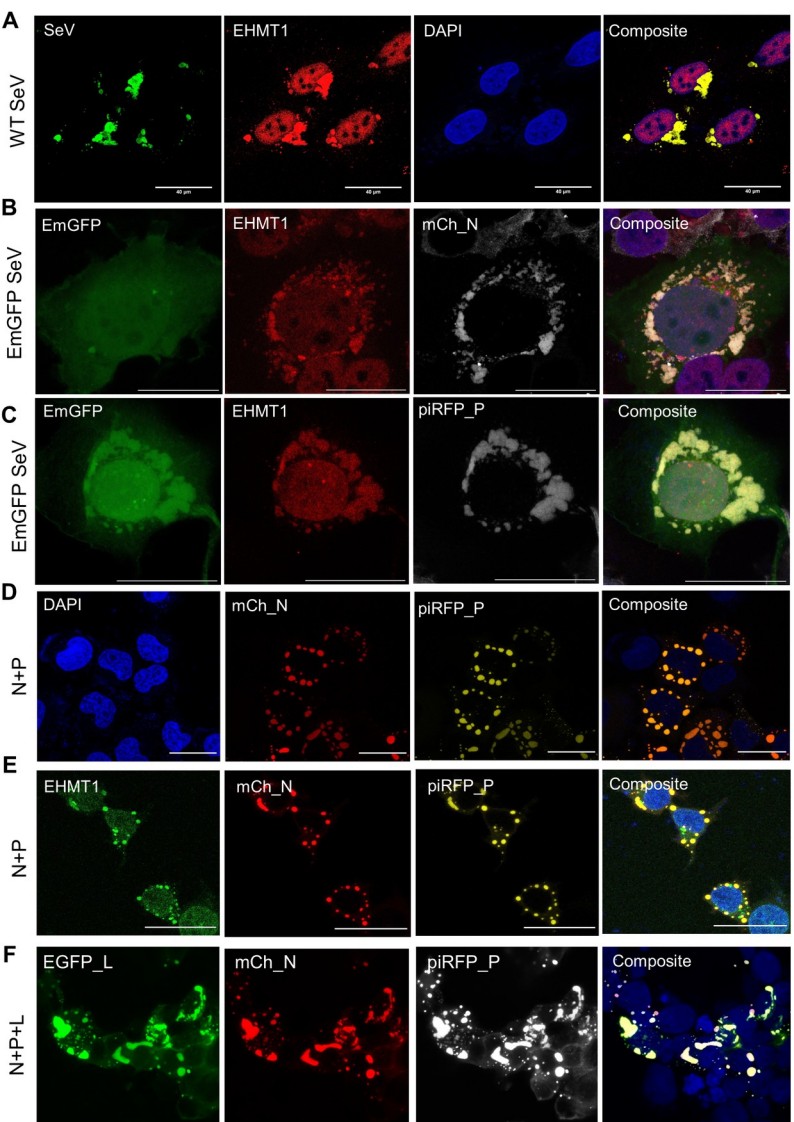

**Fig 2. EHMT1 is recruited to Sendai viral IBs.** Confocal microscopic images of (A) BEAS-2B infected with WT SeV co-immunolabelled with EHMT1 (red) and SeV (green), (B) HEK infected with EmGFP SeV (green) and transfected with mCh_N (grey), immunolabelled with EHMT1 (red), (C) HEK infected with EmGFP SeV (green), transfected with piRFP_P (grey) and immunolabelled with EHMT1 (red). (D, E) HEK co-transfected with mCh_N (red) and piRFP_P (yellow) at a ratio of 1:1 (E) immunolabelled with EHMT1 (green). (F) HEK co-transfected with Egfp_L (green), mCh_N (red), and piRFP_P (grey) in a 3:6:2 ratio. Composites of all images are with DAPI (blue) stained nuclei. Scale bar, 40 μm. Raw confocal microscopic images are deposited on BioImage Archive (Accession id: S-BIAD1362). EHMT1, euchromatic histone methyltransferase 1; IB, inclusion body; SeV, Sendai virus; WT, wild-type.

The N and P proteins of viruses belonging to the order mononegavirales form phase separated compartments in the cytosol upon their co-expression, independent of infection [28,29,36]. Although the IBs formed by co-transfection of N and P are inert structures as opposed to the IBs formed during infection, these IBs mimic the structures formed upon infection in terms of the factors recruited and their properties [28–32,36]. SeV N and P proteins upon co-transfection also formed spherical shaped IBs (Fig 2D), resembling liquid phase separated structures. We next examined if N and P are responsible and sufficient for the recruitment of EHMT1 in the absence of infection, towards which, we performed ICC against

EHMT1 in co-transfected cells. We found that EHMT1 localized to the IBs formed by N and P independent of viral infection (Fig 2E).

IBs induced by infection were heterogenous in size and shape (spherical to irregular), on the contrary, IBs formed upon co-transfection of N and P were relatively uniform in size and spherical in shape. Among the 6 proteins encoded by SeV, 3 of them, namely N, P, and L are known to facilitate viral replication [23,37]. Therefore, we speculated that since N and P were sufficient to induce IB formation, additional viral/host factors were needed to recapitulate the morphology of IBs formed during infection. Therefore, we co-expressed L protein along with N and P. Single expression of L protein exhibited diffused pattern of expression in cells expressing low levels of the protein, whereas L showed aggregate-like structures in cells expressing high levels of the protein (S2F Fig). When L was co-expressed with N and P, it co-localized to IBs (Fig 2F). Interestingly, the shape and size of IBs formed upon N+P+L transfection resembled with those formed upon viral infection. Quantification of the overall area of IBs formed upon viral infection and co-transfection revealed the formation IBs of area ranging from 0.1 to 500 $\mu m^2$ upon infection; 0.1 to 50 $\mu m^2$ upon co-transfection with N+P and 0.1 to 500 $\mu m^2$ upon triple transfection with N, P, and L. This data indicated that the expressions of all 3 viral proteins were needed to recapitulate the size and shape of SeV IBs formed during infection (S2G Fig).

To further characterise these IBs, we looked for other host proteins recruited to the IBs and found that certain host proteins like Hsp70 have been reported to interact with the viral replication proteins of mumps [38], rabies [26,39], and respiratory syncytial virus [40,41] in their IBs, where they directly influence viral replication. We then examined if Hsp70 is recruited to SeV IBs by co-immunolabelling for Hsp70 and EHMT1 in SeV infected cells, where we found that Hsp70 localized to SeV IBs (S2H and S2I Fig), which are marked by N, P, and EHMT1 proteins. ICC against Hsp70 in co-transfected cells revealed a localization of Hsp70 with the IBs (S2J Fig), indicating that the recruitment of certain host proteins is conserved with the IBs formed during infection. Altogether, we identified that EHMT1 is a part of SeV IBs, which are marked by the viral N, P, and L proteins and a host protein, Hsp70. The expression of SeV replication proteins suggests that these IBs serve as replication sites. Additionally, we demonstrated that N and P proteins are the factors responsible for association with and condensation of EHMT1 into IBs.

## A distinct nucleo-cytoplasmic form of EHMT1 associates with SeV IBs

Since the time EHMT1 was identified, it has been known to localized exclusively in the nucleus. In this study, for the first time, we observed extranuclear localization of EHMT1 into cytoplasmic SeV IBs. Considering that viral IBs recruit host proteins [30–35,38–41], we hypothesised that SeV could be inducing nuclear to cytoplasmic shuttling of EHMT1 upon infection. To test our hypothesis, we cloned the full-length EHMT1 in an mCherry_C1 vector (mCh_EH1_FL) (S3A Fig). Transfection of mCh_EH1_FL followed by confocal microscopic imaging demonstrated its localization to the nucleus (S3B Fig). We then infected the mCh_EH1_FL transfected cells with SeV to examine its cytoplasmic shuttling. Surprisingly, we did not observe any cytoplasmic mCh_EH1_FL signal in response to SeV infection (S3C Fig). While SeV IBs were formed in the cytoplasm, mCh_EH1_FL expression was restricted to the nucleus of infected cells (S3C Fig), indicating that the full-length nuclear EHMT1 does not shuttle, and the cytoplasmic protein might be different from the nuclear form. Thus, we next investigated if a unique cytoplasmic isoform of EHMT1 might be expressed in response to SeV infection. To test this, we fractionated SeV infected and uninfected cells into the cytoplasmic and nuclear fractions. As seen in Fig 3A, the nuclear fraction was enriched with LaminB1 and

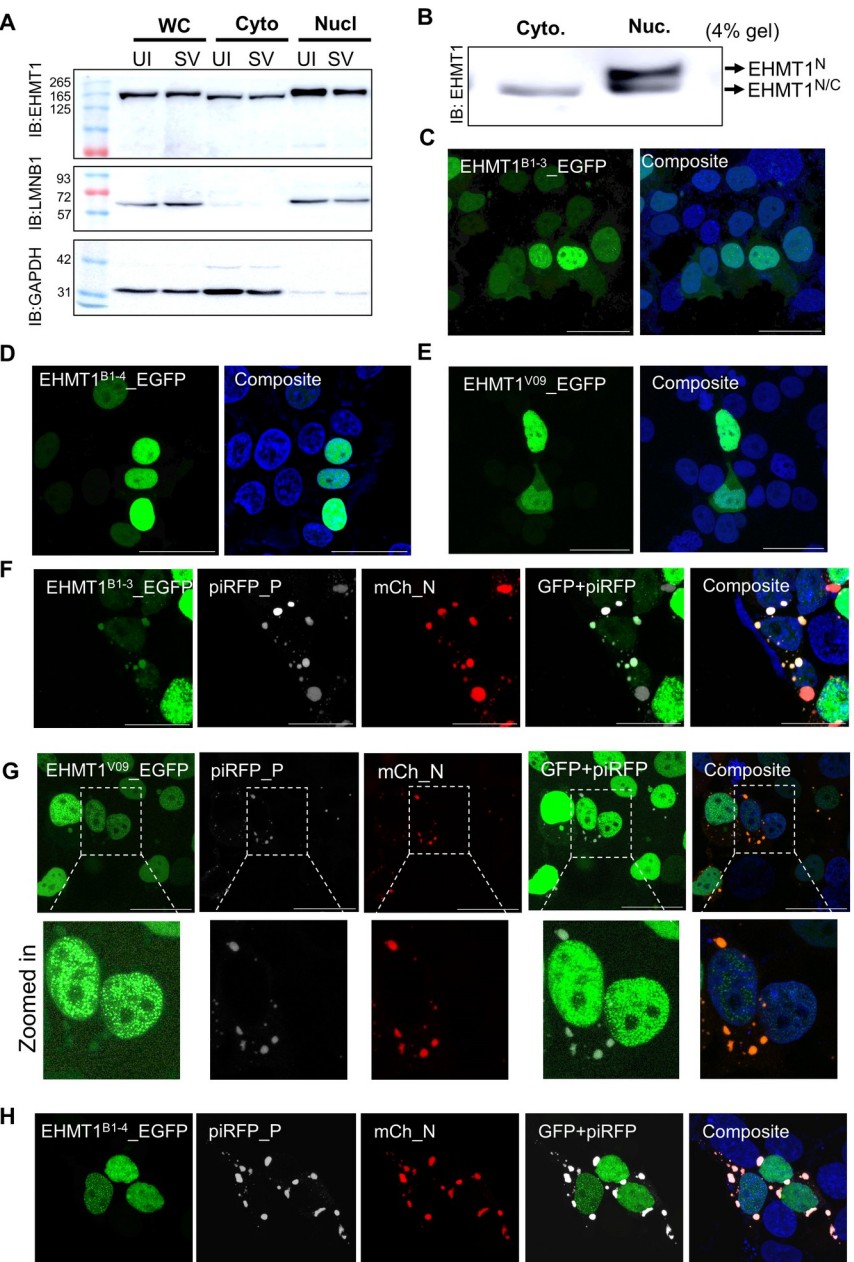

**Fig 3. A distinct nucleo-cytoplasmic form of EHMT1 associates with SeV IBs.** (A) Western blotting of the whole cell, nuclear and cytoplasmic fractions of Uninfected BEAS-2B cells (UI) and EmGFP SeV infected (SV) at 48 h p.i. Blots probed with EHMT1, LaminB1, and Gapdh. (B) Nuclear and cytoplasmic fractions of BEAS-2B cells resolved on a 4% SDS-PAGE gel, immunoblotted with EHMT1. Confocal microscopic images of HEK transfected with (C) EHMT1$^{B1-3}$_EGFP (green), (D) EHMT1$^{B1-4}$_EGFP (green), and (E) EHMT1$^{V09}$_EGFP (green). Confocal microscopic images of HEK co-transfected with mCh_N (red) and piRFP_P (grey) along with (F) EHMT1$^{B1-3}$_EGFP (green), (G) EHMT1$^{V09}$_EGFP (green), and (H) EHMT1$^{B1-4}$_EGFP (green). Composites of all images are with DAPI (blue) stained nuclei. Scale bar, 40 μm. Source data are provided as S1 Data. Raw confocal microscopic images are deposited on BioImage Archive (Accession id: S-BIAD1362). EHMT1, euchromatic histone methyltransferase 1; IB, inclusion body; SeV, Sendai virus.

cytoplasmic with GAPDH. To our surprise, we observed EHMT1 in the cytoplasmic fraction irrespective of SeV infection and the cytoplasmic EHMT1 resolved slightly lower than nuclear EHMT1 (Fig 3A). This indicated the expression of 2 forms of EHMT1, a full-length form in

the nucleus and a slightly smaller form in the cytoplasm. Since these 2 forms had a small size difference, we could not segregate them in the whole cell lysate. To sort this, we resolved the fractions on a lower percentage (4%) SDS-PAGE gel, where the nuclear EHMT1 resolved into 2 bands, the smaller band resolved at the same size as the cytoplasmic band, thereby revealing that smaller form of EHMT1 was not unique to the cytoplasm but was also expressed in the nucleus (Fig 3B). We termed this isoform as the nucleo-cytoplasmic form of EHMT1 (EHMT1$^{N/C}$). Quantification of the levels of EHMT1 in the nuclear and cytoplasmic fractions of the infected cells by normalising with the respective uninfected lanes revealed a minute but significant reduction in EHMT1 in both the nuclear as well as the cytoplasmic compartments upon infection (Figs 3A and S3D). This reduction in EHMT1 upon SeV infection was consistent with a global quantitative proteomic study performed on cells infected with SeV [42].

Compared to EHMT1$^N$, EHMT1$^{N/C}$ exhibited a small size difference; such variations are usually a resultant of PTMs like phosphorylation or due to the formation of an alternative transcript by deletion of small exons from the full-length mRNA. Upon analysis of EHMT1 transcripts deposited on Uniport, we noticed 3 isoforms of the canonical full-length form of EHMT1 (Q9H9**B1**-1298 AA long). Among them, Q9H9**B1-2** codes for the first 67 amino acids of the canonical protein; the product thus formed is unstable. Q9H9**B1-3** and Q9H9**B1-4** were missing 1185-1298aa and 809-1298aa regions, respectively. We also noticed an isoform (A0A1B0G**V09**) that was alternatively spliced with initial 93 missing bases, as a resultant of splicing events which resulted in the elimination of the first 2 exons and a part of the third exon. The 94th base had an in-frame ATG, which was the start site of this isoform; the rest of the transcript aligned 100% with the full-length form. To uncover the form of EHMT1 that exhibits cytoplasmic localization and associates with SeV IBs, we cloned all the above isoforms (except Q9H9**B1-2**) in EGFP_N1 reporter expression vector. We termed these constructs EHMT1$^{B1-3}$_EGFP; EHMT1$^{B1-4}$_EGFP and EHMT1$^{V09}$_EGFP. The above-mentioned constructs were transfected in HEK and visualized for GFP expression in nuclear versus cytoplasmic compartments. All the isoforms cloned demonstrated predominant nuclear localization signal like the canonical full-length protein (Fig 3C–3E). Interestingly, about 45% to 50% of cells transfected with EHMT1$^{B1-3}$_EGFP or EHMT1$^{V09}$_EGFP displayed a faint signal in the cytoplasm as well (Figs 3C, 3E, and S3E). However, such cytoplasmic signal was absent in cells transfected with EHMT1$^{B1-4}$_EGFP construct (Figs 3D and S3E). Upon observing cytoplasmic localization of EHMT1 isoforms, we wondered how this observation had been missed for such a long period. In immunocytochemistry experiments followed by confocal imaging, EHMT1's signal intensity is concentrated in the nuclei of cells, where the threshold intensity is corrected for the noisy faint cytoplasmic signal, which results in detection of only the nuclear EHMT1's signal. Upon deliberately setting a lower threshold intensity by correcting for the background with a negative control, we noticed a faint cytoplasmic signal of endogenous EHMT1, corroborating with the cytoplasmic band detected by western blotting as well as the localization profiles of the EHMT1 isoforms (S3F Fig).

Having identified the isoforms that demonstrate a nucleo-cytoplasmic expression profile, we next examined if the cytoplasmic form of these proteins enters SeV IBs. Towards this, we co-transfected HEK with above-mentioned isoforms, SeV N and SeV P proteins. In triple transfection experiments, we observed that EHMT1$^{B1-3}$_EGFP as well as EHMT1$^{V09}$_EGFP co-localizes with the IBs formed by N+P (Fig 3F and 3G) while EHMT1$^{B1-4}$_EGFP did not colocalize with SeV N and P proteins in the cytoplasm (Fig 3H). Quantification of the percentage of EHMT1$^{B1-3}$ or EHMT1$^{V09}$ transfected cells with colocalization into IBs revealed about 45% for EHMT1$^{V09}$ and 67% for EHMT1$^{B1-3}$ (S3G Fig). Overall, our results led to the identification of nucleo-cytoplasmic isoforms of EHMT1 that associate with SeV IBs upon infection.

Uniprot also reported 24 computationally predicted isoforms of EHMT1. A closer look at the deposited isoforms revealed that these transcripts encode for various lengths of EHMT1, encompassing either all or few domains. While it is not clear if these forms are indeed expressed in cells, we tested the localization profiles of various domains of EHMT1 into SeV IBs. EHMT1 can broadly be divided into the N-terminal half and the C-terminal half, which possesses the Ank and SET domains. Accordingly, we generated 3 constructs, one for a portion of the N-terminal, N300 (Exon 1–5), one with the activity domains, Ank+SET (Exon 13–27) and another missing a region of the N-terminus from exons 5–13, ΔEHMT1. These sequences were cloned into an EGFP_C1 reporter expression vector (S3H Fig). Upon transfection, the Ank+SET domains remained in the cytoplasm (S3I Fig) and its overexpression formed aggregation in the cytoplasm (S3I Fig). While both N300 and ΔEHMT1 showed predominant nuclear localization (S3I Fig), we observed that in few cells transfected exclusively with ΔEHMT1, a faint GFP signal was observed in the cytoplasm (S3I Fig). Intrigued by this observation, we co-transfected N300_GFP, Ank+SET_GFP, and ΔEHMT1_GFP transfected cells with SeV nucleoprotein and phosphoprotein. To our surprise, ΔEHMT1 colocalized with SeV IBs (S3J Fig) but such change in localization was not seen in N300 alone (S3J Fig). However, Ank+SET localized with SeV IBs as well (S3J Fig). This data identified the propensity of various domains of EHMT1 to associate with SeV IBs.

In summary, we found that the canonical full-length form of EHMT1 is restricted to the nucleus. The nucleo-cytoplasmic forms of EHMT1 that get recruited to SeV IBs are distinct smaller forms of the full-length protein.

## EHMTs enzymatic activity regulates SeV IB formation and viral replication

Next, we examined the requirement of EHMT's enzymatic activity in viral processes such as IB formation and replication by inhibiting the methyltransferase activity of EHMT using specific small molecule inhibitors BIX01294 [43] and UNC0642 [44]. BEAS-2B cells were treated with 3 μm BIX or 3 μm UNC and infected simultaneously with WT SeV, where IB formation was examined by immunolabelling for SeV at 16 h p.i. (Fig 4A). In untreated cells, SeV IBs were compact structures of heterogenous sizes ranging from small to large, whereas in BIX and UNC treated cells, the IBs were predominantly smaller in size (Fig 4A), indicating that EHMTs might have a role to play in the formation of large IBs. Given that IBs are sites of viral replication [26,45], which was validated in our study by demonstrating the colocalization of all the 3 components of the viral replication machinery into IBs (Fig 2F), we then assessed the impact of EHMT's inhibition on viral replication. Considering that the viral negative-sense genomic RNA as well as the mRNA have overlapping sequences, we designed primers spanning the intergenic regions of the viral genomic RNA. Since every gene has its respective start and stop codon, the sequence resulting from amplification of the intergenic region will be specific to the viral genomic RNA. qRT-PCR demonstrated about 40% to 50% reduction in viral gRNA replication upon inhibition (Fig 4B). Similarly, EmGFP SeV infected cells treated with EHMT inhibitors also demonstrated a reduction in the size of IBs (S4A Fig). Furthermore, replication of the EmGFP SeV was also reduced by about 50% (S4B Fig) at 24 h p.i. These results indicated that EHMT enzymatically facilitates the formation of large IBs, which are necessary for viral replication.

In Fig 2D and 2E, we demonstrated how co-expression of the N and P proteins were sufficient to form IBs, where EHMT1 was also recruited. However, these IBs are inert structures as compared to the ones formed upon infection, where the virus actively replicates its genomic RNA. We then tested the impact of EHMT's inhibition on these inert IBs and found that treatment with BIX/UNC did not have an impact on these IBs (S4C Fig), indicating that EHMT1

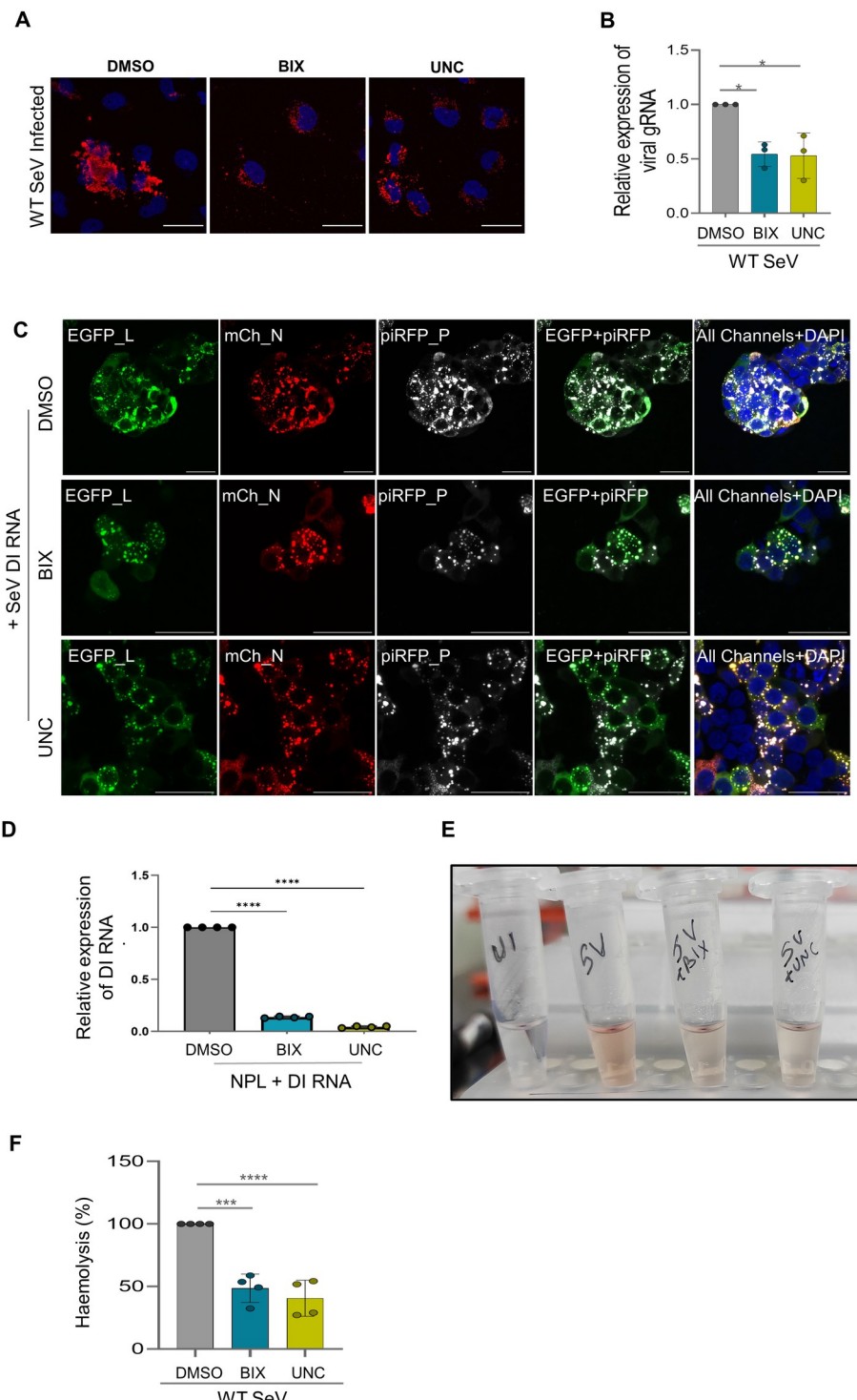

**Fig 4. EHMTs enzymatic activity regulates SeV IB formation and viral replication.** (A) Confocal microscopic composite images of BEAS-2B immunolabelled with SeV ab (red), marking the IBs in cells treated with DMSO, 3 μm BIX or 3 μm UNC and infected with WT SeV. (B) Graph plotted for the relative expression of SeV gRNA as assessed by qRT-PCR with the Ct values normalised against GAPDH. ($n$ = 3 replicates, one-way ANOVA, **, $p < 0.005$, *, $p < 0.05$) (C) Confocal microscopic images of HEK co-transfected with EGFP_L (green), mCh_N (red), and piRFP_P (grey), treated with DMSO, 3 μm BIX or 3 μm UNC and simultaneously transfected with SeV DI RNA. (D) Graph representing the relative expression of SeV DI RNA in EHMT inhibitor treated conditions, as assessed by qRT-PCR. Ct values were normalised against beta-actin. ($n$ = 4 replicates, one-way ANOVA, ****, $p < 0.0001$) (E, F) Haemolysis

assay performed on WT SeV infected cells treated with DMSO, 3 μm BIX or 3 μm UNC. (E) Representative image of tubes containing supernatant from lysed RBCs, (F) graph depicting percentage haemolysis. ($n$ = 4 replicates, one-way ANOVA, ****, $p < 0.0001$). Data from (B), (D), and (F) are mean ± SD. Source data are provided as S1 Data. Raw confocal microscopic images are deposited on BioImage Archive (Accession id: S-BIAD1362). EHMT, euchromatic histone methyltransferase; IB, inclusion body; SeV, Sendai virus; WT, wild-type.

probably exerts its effect only on large IBs formed upon infection. Quantification of the area of IBs formed (S2G Fig) had demonstrated the absence of large IBs ($>50$ μm$^2$) in the N and P co-transfection system, these IBs were able to attain large sizes of about 500 μm$^2$ upon inclusion of a third component, the polymerase (L) (S2G Fig), where they resembled the IBs formed upon infection. Having found that EHMT's enzymatic activity does not impact IBs formed by co-transfection of N and P, we then examined the effect of EHMT on large IBs formed by triple transfection (N+P+L). Inhibition of EHMT in triple transfected cells resulted in a drastic reduction in the large IBs formed (S4D Fig), thus establishing that EHMT's activity impacts the large IB population.

IBs are sites of viral genomic RNA replication; inhibition of EHMT resulted in reduction in the size of IBs, which was translated as a decrease in viral RNA replication. This could occur either due to perturbation in the sites of viral replication (IBs) or as a cascade reaction arising from fewer copies of viral gRNA, which results in reduction in viral protein synthesis, thus leading to reduced RNA replication. To decipher this process, we created a mini replicon system [46–48], where the concentration of proteins was maintained constant across various conditions and supplemented with a mimic of the viral RNA to recapitulate IBs formed upon infection. The mini replicon system was created by reconstitution of the viral replication proteins by co-transfecting cells with the mCh_N, piRFP_P, and EGFP_L plasmids in a 1:1:0.5 or 3:6:2 ratio, followed by introduction of the viral defective interfering (DI) RNA. The SeV DI RNA is about 1/10th the length of the full-length genomic RNA of SeV, known to replicate in cells and activate the immune response [23,46]. To examine if the DI RNA is successfully transfected and replicating in our system, we harvested cells at various time points and performed qRT-PCR. Comparison of DI RNA replication indicated a steep increase in replication from 12 h to 16 h, following which, the replication decreased by 24 h (S4E Fig), validating replication of the DI RNA in the mini replicon system. Treatment with EHMT inhibitors in this system indicated a drastic reduction in large IBs (Fig 4C), which was in alignment with our previous findings. Assessing the effect on DI RNA replication indicated above 80% reduction in replication in conditions where EHMT's enzymatic activity was inhibited (Fig 4D). Experiments performed in this system, where viral protein expression was controlled, led us to conclude that EHMT impacts viral replication purely by modulating the formation of replication sites/IBs.

While we found that viral IB formation was reduced upon EHMT's activity inhibition, which in turn reflected on viral replication, we further examined the implications of these processes on viral propagation. Towards this, we performed the haemolysis assay to determine the viral titre in infected cells in the presence or absence of EHMT inhibitors. The haemolysis assay is a commonly used and reliable assay for quantifying Sendai viral load [49]. To perform this assay, we infected BEAS-2B cells with WT SeV and treated them with EHMT inhibitors, BIX and UNC. The supernatant was collected 24 h p.i. and incubated with 1% RBCs on ice for attachment, followed by incubation at 37°C for lysis of RBCs. The cells were then centrifuged and supernatant containing haemoglobin from lysed RBCs were quantified; the amount of lysis being directly proportional to the viral load. From this, we observed that haemolysis was reduced by about 50% to 60% in EHMT inhibitor treated cells compared to the untreated infected cells (Fig 4E and 4F). This data further strengthened our claim that EHMT1's enzymatic activity is essential for viral replication and propagation.

In summary, we found that EHMT's enzymatic activity favours the formation of large IBs in both WT as well as recombinant SeV infected cells. This in turn reflected as a reduction in viral replication by about 50%. Inhibition of EHMT in a system where large IBs were absent indicated that EHMT does not influence the formation of inert IBs/smaller IBs. Reconstituting the system to recapitulate the large IBs formed upon infection revealed that EHMT only influences the formation of large IBs. Furthermore, the mini replicon system enabled us to conclude that perturbation of IBs (replication sites) was the reason for reduction in viral RNA replication.

## EHMT$^{N/C}$ influences SeV IB formation and viral replication via non-epigenetic mechanisms

Observation of EHMT1 in this cellular compartment, where its expression had not been reported previously led us to employ an alternative approach to specifically verify if EHMT1$^{N/C}$ is required for IB formation and viral replication. Towards this, we depleted endogenous EHMT1 by using the CRISPR/Cas9 system. A guide RNA targeting the exon3 of EHMT1 (S5A Fig) was cloned into pSpCas9(BB)-2A-GFP plasmid (EH_132), which was transfected in HEK to deplete EHMT1. The target site of the EH_132 was validated for mutation by T7 endonuclease assay (S5B Fig). We obtained about 40% to 60% reduction in the protein levels of EHMT1 (S5C Fig) in cells transfected with EH_132, where we also observed a reduction in the H3K9me2 levels (S5C Fig). We further sorted the GFP positive cells to obtain clonal populations. These cells were fractionated into the nuclear and cytoplasmic fractions, respectively, to determine the levels of EHMT1$^N$ and EHMT1$^{N/C}$. The protein was found to be significantly depleted in both the nuclear as well as the cytoplasmic fraction (S5D Fig). Cells that exhibited complete loss of EHMT1, proliferated extremely slowly and could not be propagated to establish a knockout line. Thus, we proceeded with HEK demonstrating about 70% depletion of EHMT1, infected them with WT SeV 36 h post transfection, where we immunolabelled for SeV and imaged the infected cells positive for the GFP reporter to assess IB formation (Fig 5A). We observed that the size of IBs were reduced (Fig 5A), which aligned with our observations upon inhibition of its methyltransferase activity. Assessing the SeV gRNA levels in EHMT1 depleted cells by qRT-PCR indicated a reduction of replication by about 40% (Fig 5B), indicating that the recruitment of EHMT1 to SeV IBs facilitates their formation, which in turn supports viral replication (Fig 5B).

In the above experiments, EH_132 gRNA depleted both the nuclear and cytoplasmic form of EHMT1, therefore it was difficult to assess whether the effects observed on IB formation was solely in response to the loss of cytoplasmic form of EHMT1. To dissect the roles of nuclear versus cytoplasmic EHMT1, we devised an experimental setup in which we depleted both the nuclear and cytoplasmic forms of EHMT1 by CRISPR/Cas9 (S5D Fig). Additionally, we selectively introduced the nuclear full-length EHMT1 in EHMT1 depleted cells. This resulted in a system where the nuclear EHMT1's expression was restored but the cytoplasmic EHMT1 was depleted. Towards this, we transfected EH_132 transfected cells with EHMT1 full-length construct (mCh_EHMT1_FL) to compensate for the loss of the nuclear protein (Fig 5C). These cells were then infected with WT SeV to specifically determine the effects of loss of EHMT1$^{N/C}$ on viral replication. Immunocytochemistry of SeV infected cells that were expressing gRNA (Cas9 or EH_132) and mCh_EHMT1_FL indicated that the nuclear full length EHMT1$^N$ could not rescue the defective IB formation (Fig 5D). Consequently, we observed about 50% reduction in viral gRNA replication in cytoplasmic EHMT1 depleted cells (Fig 5E). Overall, this experiment negated the role of EHMT1$^N$ on IB formation or viral replication.

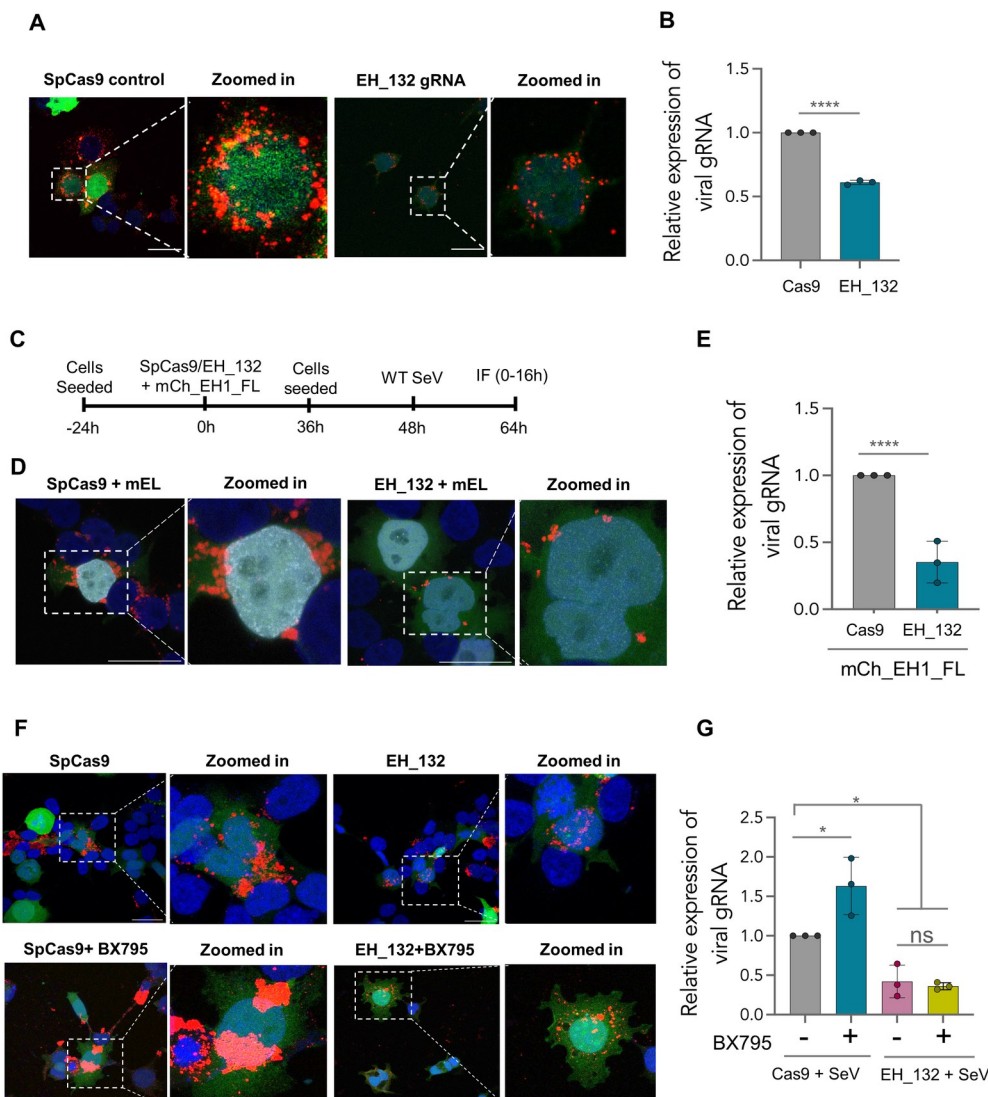

**Fig 5. EHMT$^{N/C}$ influences SeV IB formation and viral replication via non-epigenetic mechanisms.** (A, B) HEK transfected with empty Cas9 plasmid or EH_132 gRNA (GFP reporter), infected with WT SeV. (A) Confocal microscopic composite images of the cells immunolabelled with SeV ab (red). (B) Graph plotted for the relative expression of SeV gRNA normalised with GAPDH as assessed by qRT-PCR ($n = 5$ replicates, unpaired $t$ test, ****, $p < 0.0001$). (C–E) HEK transfected with SpCas9/EH_132, mCh_EH1_FL (mEL), followed by infection with WT SeV. (C) Schematic representation of the experimental procedure. (D) Confocal microscopic composite images of HEK expressing SpCas9/EH_132 (green), mCh_EH1_FL (grey), and immunolabelled with SeV ab (red). (E) Graph plotted for the relative expression of SeV gRNA normalised with GAPDH as assessed by qRT-PCR. ($n = 3$ replicates, unpaired $t$ test, $p < 0.0001$ (****)) (F, G) HEK were transfected with SpCas9 or EH_132, infected with WT SeV and simultaneously treated with BX795. (F) Confocal microscopic composite images of HEK expressing SpCas9/EH_132 (green), immunolabelled for SeV (red). Nuclei of all images are counterstained with DAPI (blue). Scale bar, 40 μm. (G) Graph plotted for the relative expression of SeV gRNA normalised with GAPDH as assessed by qRT-PCR. ($n = 3$ replicates, one-way ANOVA, *, $p < 0.05$). Data from (B), (E), and (G) are mean ± SD. Source data are provided as S1 Data. Raw confocal microscopic images are deposited on BioImage Archive (Accession id: S-BIAD1362). EHMT, euchromatic histone methyltransferase; IB, inclusion body; SeV, Sendai virus; WT, wild-type.

The activity of nuclear EHMT1 has been implicated in regulating host antiviral response by eliciting activation of the downstream IFN signalling pathway via ISGs and Ifit2 [15–17]. Down-regulation of EHMT1 either upon infection or by addition of inhibitors has been

reported to activate the Type 1 Interferon signalling pathway via ISGs and Ifit2, which imparts an antiviral response to the host by aiding in clearance of viral infection [16]. In our study, we observed that depletion/inhibition of EHMT1 resulted in reduced viral replication. Thus, to assess if the Interferon pathway regulated by the nuclear EHMT1 has any influence on the cytoplasmic viral processes, we profiled the expression of IFN and related genes upon infection. Consistent with the literature, SeV infected BEAS-2B activated expression of IFN$\alpha$, IFN$\beta$, Ifit2, IL8, and ISG20 at various levels and kinetics as indicated in S5E and S5F Fig. Transcription of genes like IRF3 and TNF$\alpha$ were not significantly altered upon infection (S5F Fig). Having found that SeV infection stimulates the expression of IFNs and its related genes, we examined the direct impact of inhibition of the IFN response in the presence or absence of EHMT1$^{N/C}$ on SeV IB formation and replication. Towards this, we created a system in which whole cellular EHMT1 was depleted via CRISPR/Cas9 and IFN$\beta$ signalling pathway was inhibited using a small molecule inhibitor, BX795. BX795 targets TBK1 kinase, which phosphorylates IRF3, thereby inducing transcription of Type1 IFN family of genes [50]. This approach allowed us to block the interferon wing controlled by the epigenetic activity of EHMT1$^{N}$ while allowing us to monitor cytoplasmic EHMT1's effect on viral replication. Cells were transfected with vector encoding for SpCas9 either without guide RNA (SpCas9) or with guide RNA targeting EHMT1 (EH_132); 36 h post transfection, cells were infected with WT SeV and simultaneously treated with BX795; 24 h post infection, cells were harvested for RNA isolation and qRT-PCR to assess the expression of various IFNs and SeV genomic RNA. From the panel of interferon related genes profiled by us, we picked IFN$\alpha$, IFN$\beta$, Ifit2, and ISG20 for further experiments. While IFN$\alpha$ levels decreased across all conditions (S5G Fig), IFN$\beta$ levels demonstrated an up-regulation by about 2-fold upon infection in SpCas9 as well as EH_132 transfected conditions, the levels of which significantly decreased in BX795 treated cells (S5H Fig). ISG20 and Ifit2 also followed a similar pattern of expression where their levels were up-regulated upon infection but treatment with the inhibitor, BX795 resulted in a reduction in their levels (S5I and S5J Fig). These results reiterated that SeV stimulates the expression of type 1 Interferon and related genes, which were reduced by treatment with the inhibitor, BX795, thus enabling us to negate any effects exerted by the Type 1 Interferon pathway on the viral processes such as IB formation and replication. In this system, we performed immunocytochemistry to examine IB formation by labelling with SeV ab (Fig 5F). We observed that in SpCas9 transfected cells treated with BX795, IBs were large (Fig 5F), these IBs seemed to be larger than the ones formed in untreated cells (Fig 5F). This indicated that inhibition of IFN pathway might enhance IB formation, based on observation of their overall morphology. Upon assessing IBs formed in EHMT1 depleted cells treated with BX795, we found that the formation of large IBs were perturbed (Fig 5F). This was in alignment with our data from Fig 5A, indicating that inhibiting the IFN pathway perturbed IB formation in conditions of EHMT1 depletion (Fig 5F). Quantification of SeV genomic RNA replication demonstrated an increase in replication in SpCas+BX795 condition (Fig 5G), which was an obvious effect considering that inhibition of the antiviral IFN pathway provides replicative advantage to the virus. However, replication of viral gRNA was significantly reduced by about 50% in EHMT1 depleted cells, either with or without inhibition of the IFN pathway (Fig 5G). No significant difference in replication levels in EHMT1 depleted cells, either treated/untreated with BX795 indicated that EHMT1$^{N/C}$ has a dominating effect over IFN inhibition. In other words, these results further indicated that in absence of EHMT1$^{N/C}$, inhibition of IFN signalling does not increase IB formation or replication.

Altogether, selective depletion of the EHMT1$^{N/C}$ as well as inhibition of the IFN pathway conclusively reinstated that it is EHMT1$^{N/C}$ and not EHMT1$^{N}$ that influences viral IB formation and replication.

## EHMT1$^{N/C}$ interacts with and methylates the SeV nucleoprotein upon infection

To investigate the mechanism by which EHMT1$^{N/C}$ regulated the size of IBs and viral replication, we studied its association with viral proteins. We began by immunoprecipitating endogenous EHMT1 from the cytoplasmic fraction of SeV infected and uninfected cells. To ensure that we are specifically immunoprecipitating the cytoplasmic EHMT1, we resolved the nuclear and cytoplasmic fractions adjacent to each other, where we found that the size of EHMT1 immunoprecipitated from the cytoplasmic fraction corresponds to the cytoplasmic band in the input lane (S6A Fig). This confirmed immunoprecipitation of the nucleo-cytoplasmic form of EHMT1. The blots when probed with SeV antibody, revealed an association with the SeV Nucleoprotein, SeV-N (57 KDa) (S6A Fig). A specific nucleoprotein band was observed only in infected samples (S6A Fig); the absence of this band from the uninfected cells (S6A Fig) indicated that the nucleoprotein band is indeed specific. To conclusively determine whether EHMT1 interacts with the nucleoprotein, we transfected cells with the SeV nucleoprotein tagged with mCherry (mCh_N), which was followed by SeV infection. EHMT1 was then immunoprecipitated from the cytoplasmic fraction of cells transfected with mCh_N and infected with SeV. The elute was resolved by western blotting and probed for mCherry, where we could detect that EHMT1 indeed associated with the nucleoprotein (S6B Fig). To further strengthen this finding and to determine any possible association of the nuclear EHMT1 with viral proteins, we transfected cells with the nuclear mCh_EH1_FL and infected with SeV. Immunoprecipitation of mCherry from the whole cell lysate of these cells revealed no association with SeV proteins (S6C Fig), thereby negating any association between the nuclear EHMT1 and the viral proteins in the cytoplasm.

Having identified 2 forms of EHMT1 (EHMT1$^{B1-3}$_EGFP and EHMT1$^{V09}$_EGFP) that co-localized with SeV IBs (Fig 3F and 3G), we tested if these nucleo-cytoplasmic isoforms interact with N protein. To verify this, we co-transfected cells with N+P along with either EHMT1$^{B1-3}$_EGFP or EHMT1$^{V09}$_EGFP and isolated the cytoplasmic fraction. Immunoprecipitation of the EHMT isoforms using GFP-specific antibody followed by western blotting using mCherry antibody identified an association of EHMT1 isoforms with the nucleoprotein (Fig 6A and 6B). To validate if these results were specific to those isoforms that localized to SeV IBs, we also performed similar experiments with EHMT1$^{B1-4}$_EGFP, which did not display nucleo-cytoplasmic expression profile or co-localize into SeV IBs (Fig 3D and 3H). Here, we did not find any association between EHMT1$^{B1-4}$_EGFP and the mCh_N (S6D Fig), indicating that the association was specific to those isoforms that co-localized to IBs. Additionally, we also performed a reverse IP to check if the viral nucleoprotein associates with EHMT1$^{N/C}$. Immunoprecipitation of mCherry from cells co-transfected with N+P and either EHMT1$^{B1-3}$_EGFP or EHMT1$^{V09}$_EGFP indicated that the viral nucleoprotein associated with both EHMT1$^{B1-3}$_EGFP as well as EHMT1$^{V09}$_EGFP (Fig 6C and 6D). These results strengthened our claim that distinct nucleo-cytoplasmic forms of EHMT1 associate with the viral nucleoprotein.

Since N+P co-transfection was sufficient to recruit EHMT1 to IBs (Fig 2E), we questioned if EHMT1 associates with N protein independent of P protein. Towards this, either N or N + P were transfected in HEK and EHMT1$^{N/C}$ was immunoprecipitated from cytoplasmic fractions. Upon probing with mCherry antibody, we identified a weak association between EHMT1$^{N/C}$ and N protein when transfected individually (Fig 6E); this interaction was much stronger in the co-transfection system, expressing both N and P (Fig 6E). This data indicated that EHMT1$^{N/C}$ predominantly associates with the nucleoprotein in a system where IBs are formed.

EHMT1 and EHMT2 are known to form functional heterodimers; since EHMT2 has a cytoplasmic isoform [10,11], we examined for any possible interaction between the 2 proteins

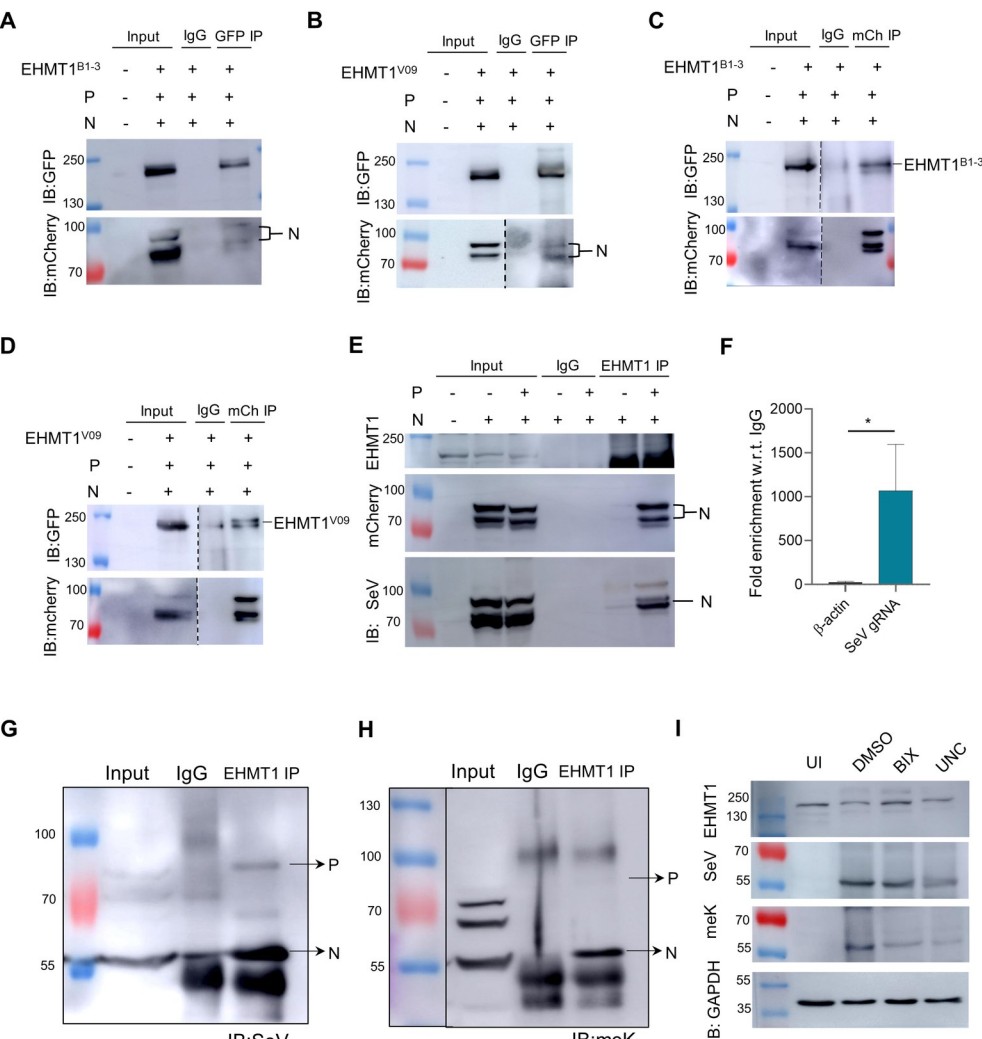

**Fig 6. EHMT1^N/C interacts with and methylates the SeV nucleoprotein upon infection.** (A, B) GFP IP from the cytoplasmic fraction of cells triple transfected with mCh_N, piRFP_P and (A) EHMT1^B1-3_EGFP or (B) EHMT1^V09_EGFP; elute western blotted and probed for GFP (top) and mCherry (bottom). (C, D) mCherry IP from cells triple transfected with mCh_N, piRFP_P and (C) EHMT1^B1-3_EGFP or (D) EHMT1^V09_EGFP, elute western blotted and probed for GFP (top) and mCherry (red). (E) EHMT1 IP from cells transfected either with mCh_N or co-transfected with mCh_N + piRFP_P, elute western blotted and probed with EHMT1, mCherry, and SeV. (F) EHMT1 RNA-IP graph representing fold enrichment of beta-actin and SeV gRNA, normalised with IgG. Data are mean ± SD. (*n* = 3, Ratio paired *t* test; *, *p* < 0.05) SeV gRNA demonstrated about 1,000-fold higher enrichment in comparison with beta-actin. (G, H) EHMT1 IP from the cytoplasmic fraction of cells infected with SeV, elute western blotted and probed with (G) SeV and (H) meK. (I) BEAS-2B cells were treated with BIX/UNC and simultaneously infected with WT SeV; cells were fractionated 24 h post infection. The cytoplasmic fraction was western blotted and probed with EHMT1, SeV, meK, and GAPDH. Source data are provided as S1 Data. EHMT1, euchromatic histone methyltransferase 1; SeV, Sendai virus; WT, wild-type.

in the cytoplasm. Towards this, we immunoprecipitated EHMT1 from the cytoplasmic fraction of uninfected and infected cells and found an association between the 2 proteins in the cytoplasm (S6E Fig). Further, we also immunoprecipitated EHMT2 from the cytoplasm of SeV infected cells and found an association with the SeV nucleoprotein (S6F Fig). This data contradicted with our immunolabelling experiments where we were unable to detect EHMT2 in SeV IBs (S1H Fig). We believe that this difference could be due to the non-natively folded state of

EHMT2 in the IBs, and/or burial of its antibody-specific epitope, due to which the antibody probably did not recognize or bind to EHMT2 in the IBs. Overall, we found that cytoplasmic EHMT1 and EHMT2 associate with each other as well as the viral nucleoprotein upon infection.

Recent studies have unravelled that IBs are sites of viral genomic RNA replication, where complexes of the N and P have been known to facilitate viral replication [26,27,34,45,51,52]. Like other members of the paramyxoviridae family, complexes of the N, P, and L have been identified to facilitate SeV replication as well [23]. Since EHMTs are associated with N and P proteins, we speculated that it could be a component of the viral RNP complex. To examine if EHMT1$^{N/C}$ is indeed associated with SeV gRNA, we immunoprecipitated EHMT1$^{N/C}$ from the cytoplasmic fraction of SeV infected cells, followed by qPCR of the immunoprecipitated elute to amplify SeV gRNA. We observed a 1,000-fold enrichment of SeV gRNA by EHMT1 IP as compared to beta-actin, which was used as a negative control (Fig 6F), indicating that EHMT1 is a part of the viral RNP complex.

Nucleo and phospho proteins, the 2 key components of the SeV RNP complex, associates with and recruits EHMT1$^{N/C}$ to IBs (Figs 2E and 6E), where its methyltransferase activity was found to facilitate the formation of large IBs (Fig 4A and 4C). These observations prompted us to question whether SeV N and P proteins could be the enzymatic substrates of EHMTs to promote the formation of large IBs. Thus, to decipher this, we examined the status of methylation of the N and P proteins in SeV infected cells. Towards this, we immunoprecipitated EHMT1 from the cytoplasmic fraction of SeV infected cells, one half of the elute was immunoblotted with SeV (Fig 6G), where we found an association with the SeV_N as well as the SeV_P proteins (Fig 6G). The other half of the elute probed with methylated lysine (meK) antibody (Fig 6H) revealed methylation of the nucleoprotein and also a weak signal of phosphoprotein (Fig 6H). The N and P proteins, which are components of the viral RNP complex, have several overlapping functions in the viral life cycle such as regulation of IB formation and viral replication. Since EHMT was found to regulate these stages of the viral life cycle, we decided to probe into the activity of EHMT on one of its potential substrates, the nucleoprotein, which is also the most abundantly expressed viral protein. To specifically determine whether EHMT is the lysine methyltransferase responsible for methylating the SeV nucleoprotein, we treated the cells with EHMT specific inhibitors BIX and UNC while simultaneously infecting them with WT SeV. The cells were fractionated 24 h p.i. to obtain the cytoplasmic fraction, which was western blotted to assess the levels of the nucleoprotein, which revealed a decrease in the levels of the nucleoprotein (Fig 6I) by about 40% to 50% (S6G Fig). We reported that inhibition of EHMT1 resulted in reduction in size of the viral IBs, which in turn reflected on a reduction in viral RNA replication (Fig 4B and 4D). This effect translates to the downstream processes like viral protein synthesis/propagation (Fig 4F), which explains a reduction in the levels of the nucleoprotein. Therefore, to assess the effect of EHMT on methylation of the nucleoprotein, we parallelly probed the blot with methylated lysine (meK) antibody, where we observed a reduction in its levels upon EHMT inhibition (Fig 6I). To specifically determine the effect of EHMT on methylation of the nucleoprotein while negating the effects of the inhibitors on the reduced viral replication, we quantified the meK bands by normalizing them with the respective nucleoprotein bands. This led to the finding that the lysine methylation of the nucleoprotein was further reduced by about 50% in EHMT inhibitor treated cells (S6H Fig), irrespective of the overall reduced levels of the nucleoprotein. Reduced levels of nucleoprotein and its methylation indicated that EHMT-mediated methylation of N protein stabilizes its expression during infection.

Finally, we tested if EHMT1's interaction with the N protein in the N+P co-transfection system leads to its methylation. Having found that N and P were responsible for the recruitment

of EHMT1 (Fig 2E), we immunoprecipitated EHMT1 from the cytoplasmic fraction of N+P co-transfected cells (S6I Fig), where we validated EHMT1's association with the nucleoprotein (S6J Fig) but found that it was unable to methylate the N protein (S6K Fig). This data thus indicated that EHMTs are the specific methyltransferases responsible for lysine methylation of the nucleoprotein upon SeV infection.

Collectively, these results reveal that distinct nucleo-cytoplasmic forms of EHMT1 associate with the viral nucleoprotein in IBs formed either upon infection or co-transfection. EHMT1 associates with EHMT2 in the cytoplasm, where EHMT2 was also found to associate with the viral nucleoprotein. Additionally, EHMT1 RNA IP indicated an association with the viral genomic RNA as well. Here, Nucleoprotein was identified as a substrate for EHMT1-mediated methylation, where EHMT1 was found to stabilise the levels of nucleoprotein.

## Incorporation of EHMT1$^{N/C}$ into SeV IBs correlates with the formation of large IB platforms and enhanced SeV replication

For cytoplasmic RNA viruses, IB formation is a dynamic process wherein small IBs coalesce with each other to form large structures to enhance viral replication [45,51]. Since the process of IB formation has not been studied for SeV, we wanted to determine the mechanism of IB formation and the contribution of EHMT1 towards formation of IBs. To address this, we monitored the behaviour of SeV IBs using live cell microscopy in HEKs transfected with mCh_N and infected with EmGFP-SeV (Fig 7A and S1 Movie). IBs formed in infected cells exhibited heterogeneity in size and shape; monitoring them over the time course of a few minutes led to the observation that spherical shaped small and intermediate sized IBs were mobile and appeared to coalesce to form larger structures (Fig 7A and S1 Movie), indicating that coalescence of smaller IBs might give rise to large IBs. Additionally, IBs also underwent fission (Fig 7A and S1 Movie), indicating a possibility of crosstalk among neighbouring IBs and the environment, a phenomenon reported to occur in IBs of several members of the order mononegavirales [27,28,51,52]. Thus, SeV IBs, like other members of the family, exhibited properties of fusion and fission.

Next, we studied the dynamics of IB formation by analysing the number and size of IBs formed at various stages of infection. Towards this, we infected BEAS-2B cells with either WT or EmGFP SeV and co-immunolabelled with EHMT1 and SeV at various time points postinfection. In WT SeV infected cells, IBs were detected as early as 3 h p.i. throughout the cytoplasm, where they appeared as tiny spherical speckles (Fig 7B). As the infection progressed, most of the spherical speckle-like IBs transformed into larger structures with irregular shape (Fig 7B). To gain clarity into the formation of IBs, we first quantified the total number of IBs formed per cell from 3 h to 72 h postinfection with WT SeV (Fig 7C). We observed that the total numbers remained constant from 3 h to 24 h p.i., followed by a steep decline post 24 h, indicating that the overall population of IBs did not significantly change until 24 h (Fig 7C). Our data on the total number of IBs was a resultant of all IBs irrespective of their heterogeneity, with the population majorly being dominated by the small IBs. Thus, the data on total number of IBs did not provide insights into the pattern of IB formation.

Studies performed on parainfluenza and metapneumovirus revealed that formation of large IBs is crucial for efficient viral replication [45,51]. In alignment with this, our results from Figs 4 and 5 also indicated that perturbation in the formation of large IBs impacted viral replication. To understand the dynamics of SeV IB formation and its relationship with replication, we first measured the size of each IB at various time points. Quantifying the size of IBs revealed that IBs ranged from 0.1 μm$^2$ to >100 μm$^2$, which led us to categorise them into 4 classes based on their size: small (0.1–1 μm$^2$), intermediate (1–10 μm$^2$), large (10–100 μm$^2$), and very

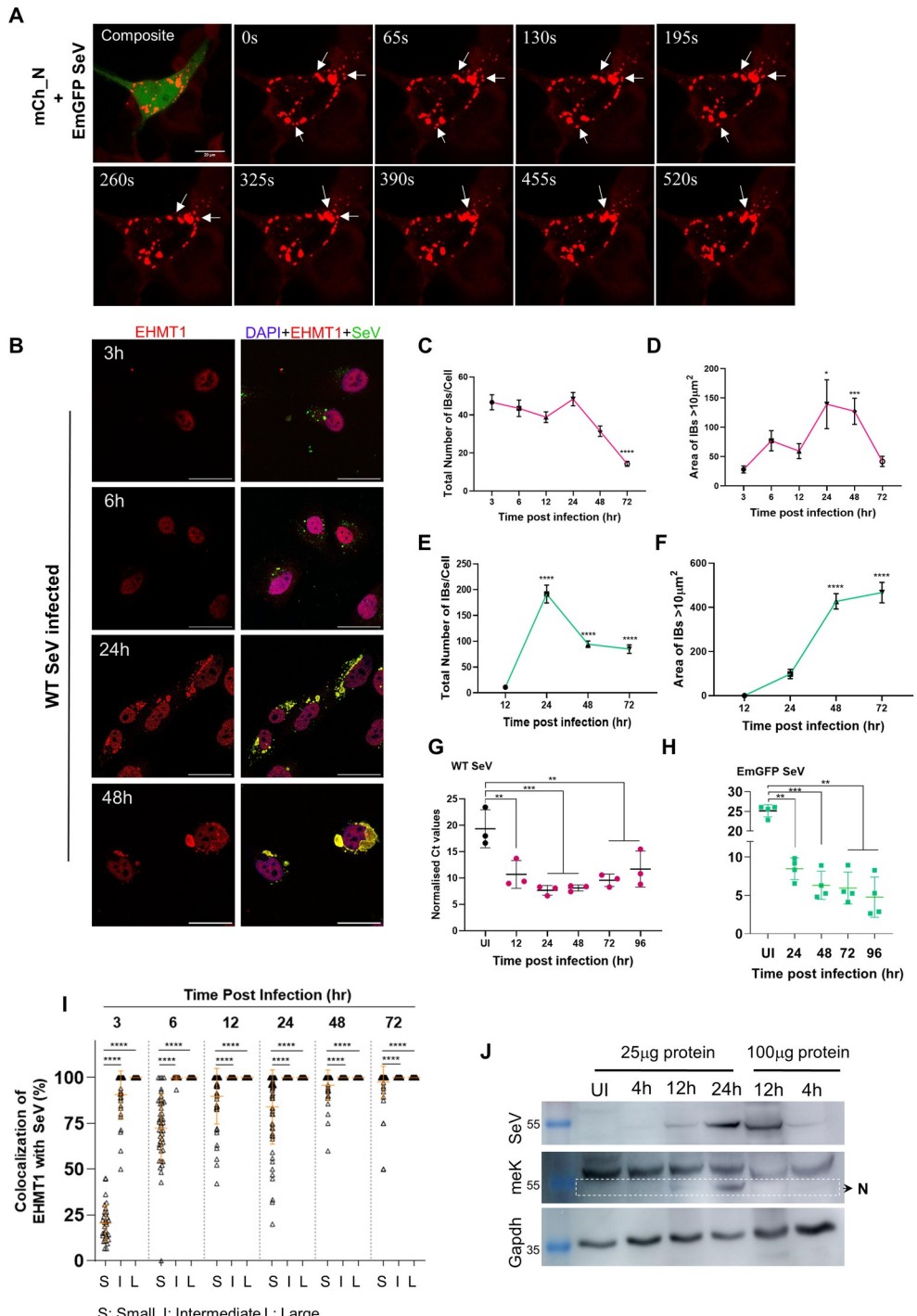

**Fig 7. Incorporation of EHMT1^{N/C} into SeV IBs correlates with the formation of large IB platforms and enhanced SeV replication.** (A) Live cell confocal microscopic images of HEK transfected with mCh_N (red) and infected with EmGFP SeV (green) at 48 h p.i.; arrows represent fusion and fission of SeV IBs. (B) Confocal microscopic images of BEAS-2B cells infected with WT SeV, fixed and co-immunolabelled for EHMT1 (red) and SeV (green) at indicated time points. (C–E) BEAS-2B cells infected with WT or EmGFP SeV were fixed and immunolabelled with SeV at various time points postinfection. Confocal microscopic images were analysed by ImageJ. (C, E) Graph representing the ±SEM of total number of IBs formed per cell for WT SeV and EmGFP SeV infected cells, respectively. (D, F) Graph representing the area ±SEM of IBs >10 µm$^2$ per cell for WT SeV and EmGFP SeV infected cells, respectively [$n > 25$ cells, Brown–Forsythe and Welch ANOVA tests]. (G, H) Replication of SeV genomic RNA at indicated time points postinfection. Graphs representing the Ct values for SeV genomic RNA normalised with GAPDH for (G) WT SeV and

(H) EmGFP SeV. Data from G and H are mean ± SD. (*n* = 4 replicates, repeated measures (RM) one-way ANOVA) (I) BEAS-2B cells were infected with WT SeV, fixed and co-immunolabelled with EHMT1 and SeV at indicated time points postinfection. Graph representing the percentage colocalization of EHMT1 with SeV in the 3 subpopulations of IBs, analysed by the Volocity Image Analysis software. Data are ±SEM. (*n* > 25 cells, ordinary one-way ANOVA) *p*-value: 0.1234 (ns), 0.0332 (*), 0.0021 (**), 0.0002 (***), <0.0001 (****). (J) Western blotting of the cytoplasmic fraction of SeV infected cells at indicated time points postinfection, immunoprobed for SeV, meK, and GAPDH. Source data are provided as S1 Data. Raw confocal microscopic images are deposited on BioImage Archive (Accession id: S-BIAD1362). EHMT1, euchromatic histone methyltransferase 1; IB, inclusion body; SeV, Sendai virus; WT, wild-type.

large (>100 μm$^2$). We observed that small IBs dominated the population from 3 h to 24 h, post which, their numbers declined at 48 h and 72 h (S7A Fig). Intermediate IBs were the second most dominant population in the above mentioned time points and demonstrated a similar trend of decline (S7A Fig). Large IBs were very few in numbers, which remained constant till 12 h; however, their population doubled by 24 h (Figs 7D and S7A), post which, there was a steep decline in their numbers. Additionally, we also observed the emergence of very large IBs at 24 h and 48 h (Figs 7D and S7A). Overall, this data indicated that while small and intermediate IBs increased in numbers slightly to attain a peak at 24 h, followed by their decline, number of the large IBs doubled by 24 h and very large IBs emerged at 24 h and 48 h. A decline in the number of small/intermediate IBs at the time point of emergence of large/very large IBs indicated coalescence of smaller IBs to form larger structures. To understand if the formation of larger IBs correlated with SeV replication, we quantified viral genomic RNA over the indicated time points (Fig 7G). A peak in SeV gRNA replication was observed around 24 to 48 h (Fig 7G), at which point, the number of large and very large IBs peaked, implying that the formation of large IBs correlated with higher replication.

For EmGFP SeV, we observed an initial delay in IB formation, where the earliest IBs were formed at 12 h p.i. (Figs 7E and S7B), followed by a rapid increase in their numbers by 24 h, with steep decline at 48 h and 72 h (Fig 7E). IBs at 12 h appeared as tiny spherical speckles (S7B Fig), which increased in numbers by 24 h (Fig 7E). The small IBs contributed to a major portion of the IB population, which was followed by intermediate and large IBs, very large IBs were occasionally observed in cells at later time points of infection. While the number of small, intermediate, and large IBs peaked at 24 h (S7C Fig), followed by a steep decline from 48 h onwards, we observed a peak in the number of very large IBs at 48 h and 72 h (Figs 7F, S7B, and S7C). This was like the trend observed with WT SeV IBs, where an increase in the number of very large IBs coincided with the point of decline of smaller IBs, indicating coalescence of smaller IBs to form larger structures as infection progressed. Quantitative analysis of the viral genomic RNA (Fig 7H) further confirmed a peak in replication at the time points of formation of very large IBs. Collectively, these data indicated an increase in the size of IBs as their numbers decreased over the time course of infection. Additionally, the appearance of very large IBs correlated with an increase in viral gRNA replication, confirming that coalescence of IBs to form large structures is essential for efficient replication.

Since inhibition/depletion of EHMT1 led to the formation of smaller IBs (Figs 4 and 5), we examined if a relationship existed between the size of IBs and the inclusion of EHMT1$^{N/C}$ over the time course of infection. To examine this, we assessed the co-localization of EHMT1$^{N/C}$ with SeV in IBs of different sizes at various time points postinfection. Object-based colocalization analysis was performed on confocal microscopic images using the Volocity Image Analysis software, where IBs were marked by SeV antibody and immunolabelled with EHMT1$^{N/C}$. At the earliest time point of WT SeV infection (3 h), we observed that about 10% of small, 85% of intermediate, and 100% large/very large IBs contained EHMT1$^{N/C}$ (Fig 7I). As the infection progressed, there was a gradual but substantial increase in incorporation of EHMT1$^{N/C}$ into

small IBs with an increase from 10% at 3 h to about 80% at 72 h (Fig 7I). Intermediate IBs exhibited about 100% colocalization by 12 h, whereas large/very large IBs exhibited 100% incorporation throughout (Fig 7I). This data demonstrated that EHMT1$^{N/C}$ is incorporated into all the large IBs; its incorporation into small/intermediate IBs increased as the infection progressed, thus demonstrating a clear relationship between the size of IBs formed during infection and the incorporation of EHMT1$^{N/C}$. Altogether, our data indicated that SeV IBs exhibit properties of fusion and fission and formed large IBs by coalescence. In addition to this, a strong correlation exists between the size of IBs and incorporation of EHMT1$^{N/C}$, indicating a possible role of EHMT1$^{N/C}$ in promoting maturation or formation of SeV IBs to generate larger platforms for efficient replication.

Since the recruitment of EHMT1 was 100% in large IBs, whose population seemed to be affected by the inhibition/depletion of EHMT1, we questioned if this reflected on the methylation of nucleoprotein. To address this, we harvested WT SeV infected cells at various time points (4 h, 12 h, and 24 h) postinfection, wherein we observed a gradual increase in the population of large IBs (Figs 7B and 7D, and S7A). Western blotting of the cytoplasmic fraction of cells harvested at various time points postinfection indicated weak expression of the nucleoprotein, the amount of which increased gradually from 12 h to 24 h (Fig 7J). Probing the blot with meK antibody revealed an increase in methylation of N from 12 to 24 h (Fig 7J). However, this observation could be attributed to an overall increase in N in a time-dependent manner. To circumvent this technical concern, we used 4 times the amount of 12 h lysate for normalization with 24 h, which gave us nearly equal amounts of N at both the time points (Fig 7J). Interestingly, although the levels of nucleoprotein were equal in both time points, the meK levels were very low at 12 h compared to 24 h (Fig 7J), indicating that methylation of nucleoprotein is a time-dependent phenomenon. Thus, an incremental recruitment of EHMT1 to SeV IBs over the time course of infection is reflected on its methyltransferase activity on the SeV nucleoprotein and formation of large IBs.

In summary, these results indicated that SeV IBs are dynamic structures that form as tiny speckles during early stages of infection. These speckles progressively enlarge over the time course of infection to form larger platforms for efficient replication of the viral gRNA. Additionally, EHMT1$^{N/C}$'s incorporation into the IBs also increased progressively, where it was completely incorporated into all the large IBs, the population impacted upon EHMT inhibition/depletion. In support of this information, we demonstrate how incorporation of EHMT1 into IBs correlates with nucleoprotein methylation and formation of large IBs.

## Methyltransferase activity of EHMT1$^{N/C}$ catalyses the growth of IBs to form larger structures

To examine if EHMTs indeed promote the maturation/growth of SeV IBs, we studied the effect of inhibition/depletion of EHMT1 on the various subpopulations of IBs. Towards this, we treated BEAS-2B cells with small molecule inhibitors of EHMT1 and 2, BIX and UNC and simultaneously infected them with WT SeV. We performed ICC against SeV to mark the IBs to determine the impact of EHMT's inhibition on their formation (Fig 4A). Analysis of the total number of IBs formed per cell (Fig 8A) demonstrated about 20% increase in the number of IBs in inhibitor treated cells. Measuring the overall size distribution of the whole IB population (Fig 8B) led to the finding that the size of IBs in untreated cells ranged from about 0.1 μm$^2$ to 500 μm$^2$, which was reduced to 0.1 μm$^2$ to 90 μm$^2$ (Fig 8B) upon EHMT1 and 2 inhibition. To understand the relationship between an overall increase in the number of total IBs and a reduction in their mean size, we studied the subpopulations of IBs. The number of small IBs demonstrated about 37% increase in BIX and 24% increase in UNC treated conditions

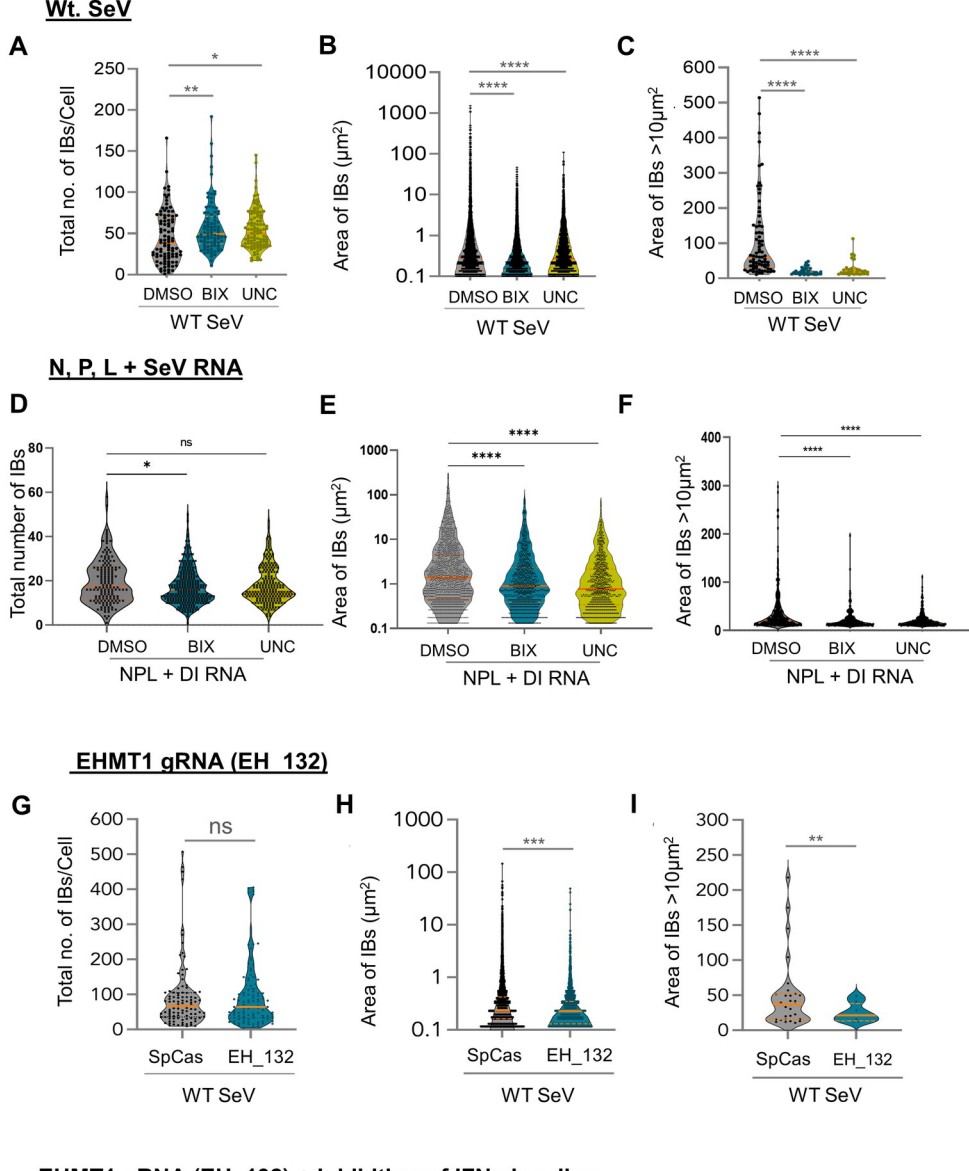

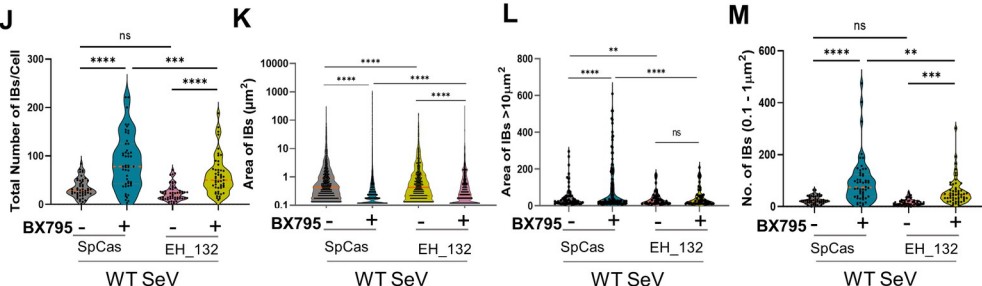

**Fig 8. Methyltransferase activity of EHMT1$^{N/C}$ catalyses the growth of IBs to form larger structures.** (A–M) Graphs representing the number and area of IBs analysed from confocal microscopic images by ImageJ analysis tool. (A–C) BEAS-2B cells were infected with WT SeV and treated with 3 μm of BIX/UNC. The cells were fixed and immunolabelled with SeV antibody 16 h p.i. Graphs representing (A) total number of IBs per cell, (B) distribution of area of the whole IB population, (C) area of IBs >10 μm$^2$. ($N$ = 3 replicates, $n$ > 100 cells, ordinary one-way ANOVA) (D–F) HEK were co-transfected with mCh_N, piRFP_P, and EGFP_L in a 3:6:2 ratio, followed by transfection with

SeV DI RNA 12 h post transfection of the plasmids and simultaneously treated with 3 μm of BIX/UNC. The cells were fixed and processed for confocal microscopy 12 h post transfection of RNA. Graphs representing (D) total number of IBs, (E) overall area of the whole IB population, and (F) area of IBs >10 μm$^2$. ($N$ = 3 replicates, $n > 100$ cells, ordinary one-way ANOVA) (G–I) HEK were transfected with empty SpCas9 or EH_132 plasmid to deplete the levels of EHMT1 and infected with WT SeV; cells were immunolabelled with SeV 16 h p.i. Graphs representing (G) total number of IBs per cell, (H) distribution of area of the whole IB population, (I) area of IBs >10 μm$^2$. ($N$ = 3 replicates, $n > 100$ cells, unpaired $t$ test), $p$-value: 0.1234 (ns), 0.0332 (*), 0.0021 (**), 0.0002 (***), <0.0001 (****). (J–M) HEK were transfected with SpCas9 or EH_132, infected with WT SeV and either untreated or simultaneously treated with BX795. Graphs representing (J) total number of IBs per cell, (K) distribution of area of the whole IB population, (L) area of IBs >10 μm$^2$ and (M) number of IBs of the 0.1–1 μm$^2$ category. ($N$ = 3 replicates, $n > 100$ cells, unpaired $t$ test between the indicated groups). Data from A to M are ±SEM. Source data are provided as S1 Data. EHMT1, euchromatic histone methyltransferase 1; IB, inclusion body; SeV, Sendai virus; WT, wild-type.

(S8A Fig), whereas intermediate IBs treated with BIX demonstrated a decrease and UNC demonstrated a slight but significant increase in numbers (S8B Fig). However, the number of large IBs declined by 77% in BIX and 70% UNC treated cells (S8C Fig), indicating that this is the most affected population. Thus, an increase in the number of small IBs contributed to the increase in total IB population. A subsequent decrease in the number of large IBs upon inhibition of the enzymatic activity of EHMT1$^{N/C}$ indicated an inhibition of coalescence/maturation. To assess the extent of inhibition of IB growth, we measured the size of large IBs (>10 μm$^2$) (Fig 8C), where we found that DMSO treated cells formed IBs with a mean area of greater than 100 μm$^2$, whereas the mean area in inhibitor treated cells was about 20 μm$^2$ (Fig 8C), indicating about 82% reduction in BIX and 75% reduction in UNC treated conditions. Thus, this data clearly indicated that EHMTs catalyse the coalescence and subsequently the growth of small IBs to form large structures.

Having established a distinction between IBs formed upon infection and independent of infection (mini replicon and N+P co-transfection) (S2G Fig), we wanted to examine whether EHMT1 regulates the formation of IBs in the co-transfection system. Towards this, we treated N+P co-transfected cells with BIX and UNC (S4C Fig) inhibitors simultaneously, the cells were imaged and IBs formed were analysed to determine their numbers and size. The total area and number of IBs per cell were very few in comparison with those formed upon SeV infection (S8D Fig); the size of these IBs were also smaller, where they mostly ranged from 0.1 to about 50 μm$^2$ (small and intermediate IBs) (S8D Fig). Very rarely, we noticed the formation of large IBs in these co-transfected cells. Treatment with BIX or UNC also did not impact the number or size of these IBs (S8D Fig). These results are consistent with lack of N protein methylation in N+P transfected cells, where EHMT1 did not seem to affect the size of IBs (S6K Fig).

When we performed similar experiments of EHMT inhibition in the mini-replicon system, we observed a minute but significant reduction in the total number of IBs in BIX treated condition, but this reduction was not significant with UNC (Fig 8D). We also noticed that overall size of the IBs was decreased in EHMT1 inhibitor treated cells (Fig 8E) and noticeably IBs over 10 μm$^2$ in area were significantly reduced (Fig 8F). Analysis of subpopulations of IBs did not demonstrate significant changes in small or intermediate IBs (S8E and S8F Fig) but number of large IBs (>10 μm$^2$) were significantly reduced (S8G Fig). The mini-replicon system allowed us to provide the system with constant amounts of the viral proteins and DI RNA, observation of reduced large IBs in this system can be attributed to the inhibition of EHMT1. These results further reiterated the significance of EHMT1 in IB coalescence to form larger structures.

Having studied the influence of EHMT's enzymatic activity on IB formation, we further studied the effects of EHMT1 depletion on the formation of IBs in cells infected with SeV. Towards this, CRISPR/Cas9 targeting of EHMT1 was employed; EH_132 plasmid (with GFP reporter) was transfected in HEK to deplete the levels of EHMT1. GFP positive infected cells were imaged from both SpCas9 empty vector and EH_132 transfected cells (Fig 5A). By

analysing the IBs marked by SeV antibody at 16 h p.i., we observed no significant change in the total number of IBs (Fig 8G) but there was a minute decline in the mean size of IBs (Fig 8H) formed upon depletion of EHMT1. Upon classification of IBs into their subpopulations, the mean number of small IBs were found to increase by 10% (S8H Fig), whereas intermediate IBs displayed a 52% reduction in numbers (S8I Fig) and large IBs reduced by 76% (S8J Fig) in EHMT1 depleted population. Analysing the area of large IBs formed revealed about 60% decrease in the size of the largest IBs formed upon depletion of EHMT1 (Fig 8I). We believe that the subtle differences in the increase in small IBs and decrease in large IBs compared to that of the inhibitor could be due to about 40% EHMT1 present in the system (S5C Fig).

Since EH_132 targeted both nuclear and cytoplasmic EHMT1, we wanted to specifically determine the effects of cytoplasmic EHMT1 on IB formation. To address this, we transfected EH_132 transfected cells with mCh_EHMT1_FL (Fig 5C) to compensate for the activity of the nuclear EHMT1. These cells were then infected with the WT SeV and immunolabelled with SeV antibody to assess IB formation (Fig 5D). Quantification of the IBs thus formed revealed no significant increase in the total number of IBs (S8K Fig) but a decrease by about 50% in the area of IBs $>10$ μm$^2$ (S8L Fig). These results aligned with our observations on cells depleted of total EHMT1, thus indicating that the effect of EHMT1 observed on IB formation is a resultant of the cytoplasmic EHMT1$^{N/C}$.

Next, we also assessed the contribution of the host defence system regulated by epigenetic activity of nuclear EHMT1 versus non-epigenetic role of cytoplasmic EHMT1 in regulating IB sizes. Towards this, we transfected cells with SpCas9/EH_132, treated them with BX795 36h post transfection and simultaneously infected with SeV. These cells were fixed and processed for immunolabelling with SeV at 24 h p.i. (Fig 5F). In alignment with our data from Fig 8G, we observed no significant change in the total number of IBs in EHMT1 depleted cells (Fig 8J), whereas, the numbers increased upon BX795 treatment (Fig 8J), either with or without depletion of EHMT1 (Fig 8J). However, the total numbers were reduced in EH_132 + BX_795 as compared with SpCas9 + BX_795 condition (Fig 8J). This indicated that the inhibition of the IFN pathway favoured the formation of IBs. The overall area of IBs was higher in SpCas9 +BX795 condition when compared to SpCas9 alone (Fig 8K); however, treatment with BX975 did not lead to an increase the area in EH_132 condition (Fig 8K). Analysis of the large IBs revealed a significant increase in the numbers and area in SpCas+BX_795 condition as compared with SpCas9 (Figs 8L and S8M). In contrast, the large IBs were significantly impacted both in EH_132 and EH_132 + BX975 conditions (Figs 8L and S8M). Further in-depth analysis identified an increase in small IBs upon BX975 treatment in SpCas9 and EH_132 (Fig 8M) albeit at different levels, indicating that inhibition of the IFN signalling pathway promotes accumulation of small IBs. Perturbed formation of large IBs in EH_132 conditions (Figs 8L and S8M) indicated that absence of cytoplasmic EHMT1 impairs coalescence even upon inhibition of IFN signalling. Taken together, these results demonstrated that while inhibition of the antiviral IFN pathway favoured the virus by promoting the formation of small IB, coalescence of IBs to form large structures was purely governed by the enzymatic activity of EHMT1$^{N/C}$.

These results were in alignment with our previous data where we claim that EHMT1$^{N/C}$ influences the formation of large IBs. Here, we demonstrate that despite inhibition of the IFN pathway, large IBs were affected in EHMT1 depleted condition, indicating that it is not the nuclear but the cytoplasmic EHMT1 that influences SeV IB formation and viral replication in the cytoplasm of the host. Altogether, our data on EHMT1$^{N/C}$ incorporation into IBs (Fig 7I), its enzymatic inhibition and depletion (Figs 4 and 5) convincingly demonstrate that the

methyltransferase activity of EHMT1$^{N/C}$ is essential for formation of large IBs, which is critical for efficient replication.

## Cytoplasmic condensation of EHMT1 is conserved among other members of the single-strand RNA virus family

Inclusion body formation is a shared feature among RNA viruses [26–29,52]. Therefore, we tested if the condensation of EHMT1 is a common response to RNA viral infections [Ex: Chandipura (CHPV) and Dengue (DenV)]. CHPV is a negative-strand RNA virus and an emerging tropical pathogen which forms cytoplasmic IBs [53]. MEFs were infected with CHPV and immunolabelled with EHMT1 and CHPV L protein at 6 h and 8 h p.i. CHPV condensates labelled with CHPV_L antibody indicated that CHPV IBs did not resemble SeV IBs but had a morphology of spherical speckle-like structures (Fig 9A) as described previously and were composed of heterogenous sizes [53]. Interestingly, EHMT1 also formed condensates in the cytoplasm (Fig 9A), where it was found to co-localize specifically with large IBs labelled with CHPV_L (Fig 9A).

We also employed DenV, a single-strand positive-sense RNA virus and a member of the Flaviviridae family. The DenV Non-Structural protein 1 (NSP1) is a part of the viral replication complex, also a marker of DenV replication organelles, known as Vesicle packets [54,55]. To examine if EHMT1 condensates in response to DenV infection, HepG2 cells infected with Dengue virus were immunolabelled with EHMT1 and Nsp1 parallelly at 24 h and 48 h postinfection. DenV infected cells marked with Nsp1 demonstrated a reticulate pattern of staining (invaginations from the endoplasmic reticulum) along with a few punctate-like structures (Fig 9B) as demonstrated previously [55,56]. Infected cells labelled with EHMT1 also demonstrated a similar punctate pattern of staining in the cytoplasm (Fig 9C), which partly resembled the pattern of Nsp1 staining. EHMT1 as well as NSP1 antibodies were raised in rabbit; therefore, we were unable to perform co-immunolabelling. Nonetheless, our data suggests that the formation of EHMT1 condensates is not a unique phenomenon restricted to Sendai virus infection but may be a broad response to multiple types of viral infections.

## Discussion

RNA viruses are obligate parasites that require the host cellular machinery to complete their life cycle. These viruses trigger profound changes in the gene expression of specific host proteins and bring about global reorganization of the host proteome to facilitate their replication [36,40,41,57]. Notably, infection by these viruses induce the formation of biomolecular condensates called inclusion bodies [26–29]. Depending on the stage of infection, the size, shape, and number of IBs vary in an infected cell [28,29,45]. Formation of IBs is initiated by aggregation of newly synthesized IDR-rich viral nucleo of phospho proteins [28,36]. During early phase, IBs are small with minimal viral proteins and their size increases as the infection progresses. Larger IBs represent mature IBs which have been shown to contain host factors including chaperones [26,30–34,38–41,57,58]. Given that different steps of viral life cycle occur within IBs, the biogenesis and maturation of these structures are crucial for viral replication [58,59]. Understanding these processes may lead to understanding pathogenesis of these viruses and designing therapeutic interventions.

Formation of such distinct, de novo cytoplasmic organelles is regulated by the concentration of proteins and nucleic acids, chaperones, and PTMs [60–65]. While the concentration and composition of biomolecules has been studied by in vitro assays, deciphering how PTMs influence IBs has been challenging due to the technical difficulties involved in isolating these structures sensitive to ionic strength and detergents [65–68]. Nonetheless, few PTMs like

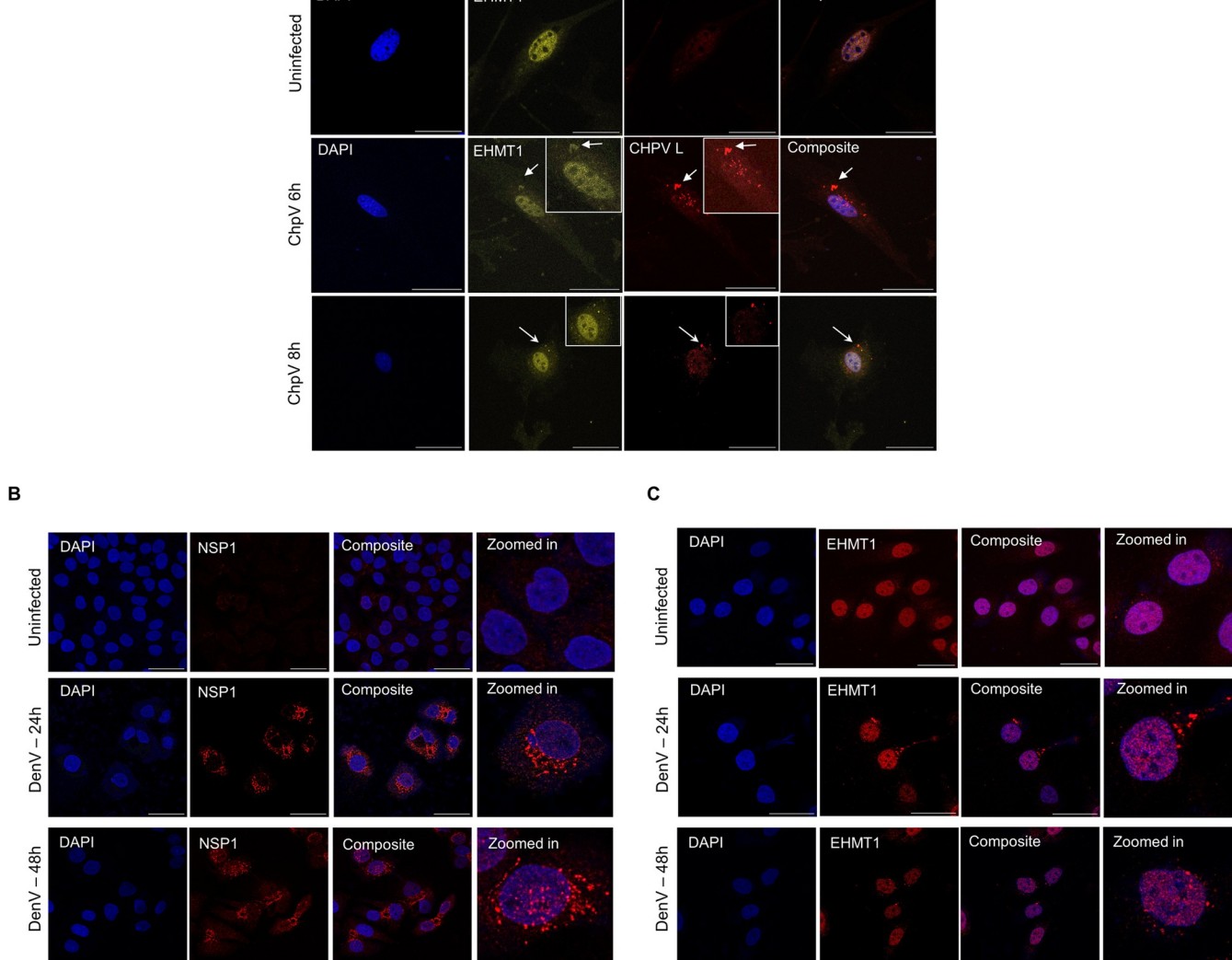

**Fig 9. Cytoplasmic condensation of EHMT1 is conserved among other members of the single-strand RNA virus family.** Confocal microscopic images of (A) MEF uninfected or infected with Chandipura virus (ChpV) at 6 h and 8 h p.i., immunolabelled with EHMT1 (yellow) and ChpV_L (red). Arrows indicate cytoplasmic condensates of EHMT1 colocalizing with ChpV IBs marked by the L protein. (B, C) HepG2 cells uninfected or infected with Dengue virus, immunolabelled with (B) Nsp1 (red) or (C) EHMT1 (red) at 24 h and 48 h p.i. Composite of all images are with DAPI (blue) stained nuclei. Scale bar, 40 μm. Raw confocal microscopic images are deposited on BioImage Archive (Accession id: S-BIAD1362). EHMT1, euchromatic histone methyltransferase 1; IB, inclusion body; MEF, mouse embryonic fibroblast.

SUMOylation, arginine methylation, phosphorylation, and ubiquitylation have been reported to directly influence biomolecular condensates but a vast majority of them remains to be studied [66,67,69–73]. In this regard, our study identified a protein lysine methyltransferase, EHMT1$^{N/C}$ as a proviral host factor for SeV pathogenesis and provides insights into how SeV engages with EHMT1$^{N/C}$ for successful replication. We report that upon SeV infection, EHMT1$^{N/C}$ localizes to viral cytoplasmic condensates as early as 3 h. Among the 6 SeV proteins, EHMT1$^{N/C}$ associates with the nucleoprotein and phosphoproteins and their co-expression in cells was sufficient for the condensation and recruitment of EHMT1$^{N/C}$. At the functional level, localization of EHMT1$^{N/C}$ into viral IBs leads to formation of larger IBs and

efficient SeV replication. Consistently, we observed small IBs and reduced replication upon knockdown or inhibition of EHMT1$^{N/C}$.

The Nucleoprotein is a critical multifunctional protein, which protects the viral RNA, interacts with and recruits host components to IBs, thereby aiding in IB formation and enabling viral genomic RNA replication within the IBs, among its several other functions [26–35]. EHMTs were found to methylate the lysine residue/s of the viral Nucleoprotein upon infection, inhibition of which led to reduction in levels of the Nucleoprotein. Since the viral Nucleoprotein has been implicated in almost every stage of the viral life cycle and EHMTs regulated its levels via methylation, it is prudent to believe that these enzymes facilitate viral replication. In combination with methylation of N protein, EHMTs might have additional host-specific substrates that aid in viral replication which still needs to be identified. Overall, our study has explored an underappreciated role of a host methyltransferase in viral pathogenesis. Identification of such protein–protein interaction is not only critical for SeV biology but can provide mechanistic insights into possibly conserved mechanism of host–pathogen interaction networks of other members of the order mononegavirales. As such, in the current study, we demonstrate that EHMT1 condensation in IBs is not unique to SeV but is a shared property among at least two pathogenic viruses studied.

Posttranslational modifications (PTMs) on proteins generate diversity of a protein pool [74] and are responsible for regulating a large number of biological processes in both host and pathogen [75–79]. RNA viruses encode for a handful of proteins that are multifunctional in nature. Thus, harnessing host enzymes responsible for PTMs to impart diversity to their proteins, thereby increasing the affinity of protein: protein interactions can be a powerful strategy for viruses to establish efficient replication. In support of this phenomenon, we found that EHMT1$^{N/C}$ methylates the SeV Nucleoprotein. Association of EHMT1$^{N/C}$ with N and methylation of N increased as the infection progressed. The timeline of increased methylation of N protein coincided with the formation of larger IBs and increased replication of SeV. Consistent with these mechanistic insights, we observed that inhibiting the catalytic activity of EHMT1$^{N/C}$ using small molecule inhibitors highly reduced the abundance of larger IBs and decreased SeV replication. Prior to our study, the only report which demonstrated involvement of a lysine methyltransferases in RNA viral replication organelles is by Chen and colleagues, wherein Smyd3 facilitated transcription of Ebola virus [35]. However, whether Smyd3 mediates methylation of viral proteins is unknown. In the current study, we for the first time demonstrate lysine methylation as a PTM to regulate the biomolecular condensation in the context of host–pathogen interaction.

Viruses are known to recruit enzymes such as kinases to phosphorylate several of their proteins. For example, human T-cell lymphotropic virus protein Tax [80], VP30 of Ebola [81], Nucleoprotein of Influenza [77], etc. are known to get phosphorylated, which eventually aids in activating viral gene transcription. A series of biochemical studies showed that GCN5 and PCAF were responsible for acetylation of the nucleoprotein in Influenza virus [82]. In addition, studies on viral envelope proteins identified its glycosylation to facilitate viral entry to dampen the immune response of the host [83]. To our knowledge, this is the first study where we report the involvement of a lysine methyltransferase and its methyltransferase activity being directly utilized by ssRNA virus in the formation/maturation of IBs (or Replication Organelles) for efficient replication.

In a high-throughput study performed to determine the enzymatic substrates of EHMT1 [6] several extranuclear proteins, including several mitochondrial, ER and cell membrane specific proteins were detected [6]. These observations were puzzling given the nuclear subcellular localization EHMT1 [8]. Our findings of a distinct isoform of EHMT1 (EHMT1$^{N/C}$), can be instrumental in studying EHMT1$^{N/C}$ mediated non-histone, non-nuclear protein regulations

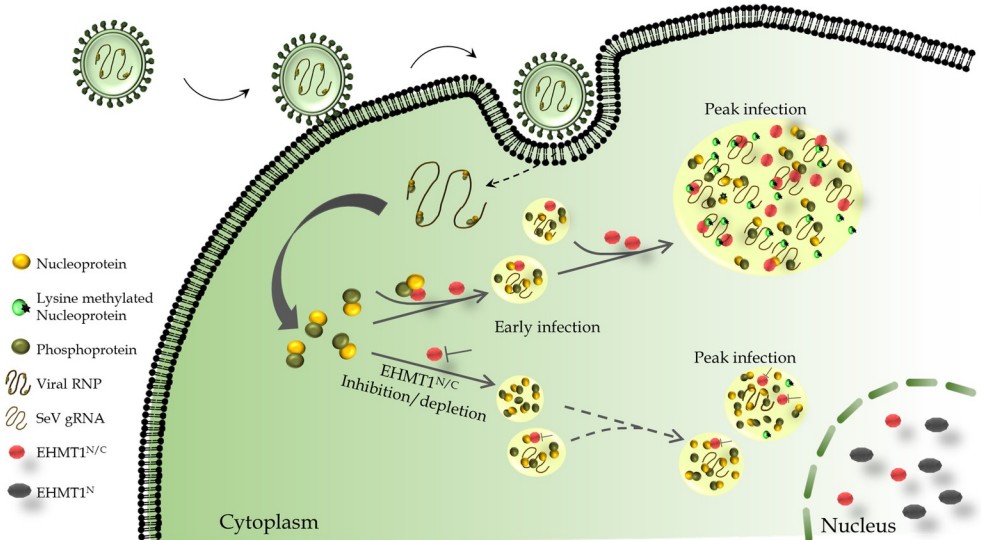

**Fig 10. Model demonstrating distinct Nucleo-cytoplasmic form of EHMT1, recruited by the virus to IBs, facilitates IB coalescence by methylation of the Nucleoprotien.** Upon SeV infection, the cytoplasmic form of EHMT1, EHMT1$^{N/C}$ is recruited to SeV IBs by the N and P proteins in a time-dependent manner as infection progresses. EHMT1$^{N/C}$ plays an enzymatically active role in coalescence of IBs by methylating N, leading to the formation of large IBs and efficient viral replication. Inhibition of EHMTs activity or depletion of EHMT1$^{N/C}$ leads to impairment in coalescence of IBs. EHMT, euchromatic histone methyltransferase; IB, inclusion body; SeV, Sendai virus.

and their complex interactomes. While we have identified how EHMT1$^{N/C}$ plays a significant role in RNA viral infection, its expression in the cytoplasm independent of infection in multiple cell lines indicates its potential role in regulating several biological processes. Knockout of EHMT1 in mice results in embryonic lethality [1], haploinsufficiency manifests as a neurodevelopmental disease called Kleefstra Syndrome [84], Copy Number Variations were reported in Schizophrenia [85], and elevated expression was found in several cancers [86,87] and Alzheimer's Disease [88]. Until now, our understanding about these diseases emanated from the regulation of processes governed by nuclear EHMT1 but the identification of a novel nucleo-cytoplasmic form extends beyond viral pathogenesis and is expected to enhance our understanding about the role of EHMT1 in development and disease.

In conclusion, our findings discovered previously uncharacterised nucleo-cytoplamic forms of EHMT1 and their role as proviral factors at the interface of host–virus interactions. At mechanistic level, our work demonstrated lysine methylation as a posttranslation modification of SeV N protein to build large IBs which are necessary for efficient replication (Fig 10). These findings will not only open new avenues to address questions pertaining to host–pathogens interaction and therapeutic interventions but will also be instrumental in illuminating unknown functions of EHMT1$^{N/C}$ in the context of development and disease.

## Materials and methods

### Ethics statement

Human blood was collected according to the approved protocols from the Institutional Human Ethical Committee (inStem/IEC-10/003) of the Institute for Stem Cell Science and Regenerative Medicine. The participants were informed, and an informed consent was received.

## Cell culture

Neonatal human dermal fibroblasts (HDFs) were purchased from ScienCell (2310), MEFs were isolated from mouse embryos 13 d.p.c. Fibroblasts, HEK 293, BEAS-2B, and HepG2 cell lines were cultured in Dulbecco's Modified Essential Media (DMEM, high glucose, GlutaMAX (Gibco)), supplemented with 10% (v/v) heat-inactivated fetal bovine serum (Gibco), and 1X Non-Essential Amino Acids (Gibco). The cells were incubated in a 37˚C, 5% CO2 incubator. For passaging, cells were briefly treated with 1X TrypLE (Gibco).

## Viruses

Institutional biosafety approval was obtained from CSIR-IGIB. Cytotune OSKM Sendai virus (A16517, Invitrogen) and EmGFP reporter Sendai Virus (A16519, Invitrogen) were part of the Cytotune -iPS 2.0 Sendai Reprogramming kit and EmGFP Sendai Fluorescence Reporter kit. Wild Type Sendai Virus (Z strain) was propagated in the allantoic sacs of 10-day-old embryonated chicken eggs. Cells were infected with an MOI of 2. Human patient derived Chandipura Virus (strain number 1653514) was propagated in Vero cells. Dengue virus (Type II) was isolated from human patient serum and propagated in Vero cells.

## Immunocytochemistry

Cells were seeded on glass coverslips of 0.167 mm thickness at 40% confluency, and 24 h post seeding, they were either transfected or infected; the cells were then fixed with 4% paraformaldehyde (w/v) for 10 min at RT. This was followed by 3 PBS washes and permeabilization with 0.5% TritonX-100 (v/v) in 1% BSA for 10 min at RT. The cells were then washed with PBS and blocked with 5% BSA for 1 h at RT. Primary antibodies diluted in 5% BSA at recommended concentrations were incubated overnight at 4˚C. The cells were then washed with PBS thrice and the respective fluorophore conjugated alexa secondary antibodies diluted in 5% BSA were incubated for 40 min at RT. After PBS washes, DAPI was added for 5 min at RT. The coverslips were then washed with PBS, air-dried, and mounted on glass slides with Vectasheild mounting media.

For co-immunolabelling with EHMT1 and SeV, we sequentially incubated the cells first with Rb EHMT1 (Novus) antibody at 1:100 dilution overnight at 4˚C, Chk SeV (abcam) antibody was added on the following day at 1:2,500 dilution. The samples were incubated overnight at 4˚C, followed by PBS washes. The alexa secondary fluorophore-conjugated antibodies were mixed at a dilution of goat-anti-rabbit 647 (1:200) and goat-anti-chicken 568 (1:1,000). The rest of the protocol followed is as explained in the previous paragraph.

## Confocal microscopy and image analysis

Confocal microscopic imaging was performed for the samples either on Olympus FV3000, Leica SP8 or Nikon A1R confocal microscope. Representative images were acquired from several fields in each sample. Images were extracted, pseudo colours were assigned based on the secondary antibody, dye or label used; z-stacks of the maximum intensity projection were constructed using ImageJ.

Live cell imaging was performed on cells seeded in coverslip bottom confocal dishes using the 63× oil immersion objective on the Leica SP8 confocal microscope.

Quantitation of the number and size of IBs was performed on ImageJ, for which Z-stacked images were converted to 8-bit and greyscale. The threshold was adjusted to eliminate the background and an ROI was marked for each cell to measure the area of each IB within a cell.

For object-based co-localization analysis, Volocity Image Analysis software from Perkin Elmer was used. Z-stacked raw images were processed for noise reduction on the Leica LAX software before importing to Volocity. The populations were defined based on the channels for SeV and EHMT1 and an ROI was marked for each cell, from which the nucleus was excluded. Object size guide was set at 25 $\mu m^3$ to separate touching objects and the colocalization was measured.

## Cloning and transfection

The SeV Nucleoprotein, Phosphoprotein, and RdRP (or L) were amplified from the cDNA of EmGFP SeV-infected cells. The Nucleoprotein was inserted into the MCS of mCherry_C1 plasmid (#632524, Addgene) at the ECoR1 and BamH1 sites using the SeV Nucleoprotein primers. SeV Phosphoprotein was inserted into the MCS of piRFP670-N1 (#45457, Addgene) at the BglII and KpnI sites using the SeV Phosphoprotein primers. SeV RdRP (L) protein was inserted into the MCS of pEGFP-N1 plasmid (#6085–1, Addgene) at the SalI and SacII sites using SeV_L specific primers. EHMT1 full-length was subcloned from the V5 tagged EHMT1 plasmid into the MCS of mCherry_C1 plasmid at the BglII and KpnI sites using the mCh_EHMT1_FL primers. EHMT1[B1-3], EHMT1[B1-4], and EHMT1[V09] were subcloned from the V5 tagged EHMT1 plasmid into the MCS of pEGFP-N1 plasmid (#6085–1, Addgene) at the BglII and KpnI sites using isoform specific primers. N300, Ank+SET, and Delta EHMT1 were amplified from cDNA of HEK and inserted into the MCS of pEGFP-C1 plasmid (Addgene). EHMT1 guide RNA was targeted against the exon3 of EHMT1 gene; it was cloned into the pSpCas9(BB)-2A-GFP (PX458) (#48138, Addgene). Primer sequences used for cloning are provided in Table 1. For episomal reprogramming, pCXLE-hSK (#27078), pCXLE-hOCT3/4-shp53-F (#27077), and pCXLE-hUL (#27080) plasmids were a gift from Shinya Yamanaka [89].

For the nuclear EHMT1 compensation experiments, HEKs at 40% confluency were transfected simultaneously with EH_132/SpCas9 and mCh_EH1_FL plasmids, and 36 h post transfection, the cells were trypsinised and seeded for infection at 48 h. The cells were then harvested 16 h post infection with the WT SeV.

**Table 1. Primers used for cloning.**

| Primer name | Primer sequence |
|---|---|
| SeV_Nucleoprotein (Forward) | 5′ TCACTCGAATTCCATGGCCGGGTTGTTGAG 3′ |
| SeV_Nucleoprotein (Reverse) | 5′ TAAGCAGGATCCCTAGATTCCTCCTACCCCAGCTA 3′ |
| SeV_Phosphoprotein (Forward) | 5′ TCACTCAGATCTGCCACCATGGATCAAGATGCCTTCA 3′ |
| SeV_Phosphoprotein (Reverse) | 5′ TAAGCAGGTACCTCGTTGGTCAGTGACTCTATGTC 3′ |
| SeV_RdRP (L) (Forward) | 5′ TCACTCGTCGACGCCACCATGGACGGGCAAGAGTCCTC 3′ |
| SeV_RdRP (L) (Reverse) | 5′ TAAGCACCGCGGTCCAAGCTGTCATATGGCTCGAT 3′ |
| mCh_EHMT1_FL (Forward) | 5′ TCACTCAGATCTATGGCCGCCGCCGATGCCGA 3′ |
| mCh_EHMT1_FL (Reverse) | 5′ TAAGCAGGTACCTCATAGGGGGTCGGCGGCAGCC 3′ |
| EH_132 (Forward) | 5′ CACCGCGCCGACGTCAAGGTCCACA 3′ |
| EH_132 (Reverse) | 5′ AAACTGTGGACCTTGACGTCGGCGC 3′ |
| EHMT1[B1-3]_EGFP (Forward) | 5′ TCACTCAGATCTGCCACCATGGCCGCCGCTGAT 3′ |
| EHMT1[B1-3]_EGFP (Reverse) | 5′ TAAGCAGGTACCTCAGAGGTAAGAGTCCTCTTCCCGAA 3′ |
| EHMT1[B1-4]_EGFP (Forward) | 5′ TCACTCAGATCTGCCACCATGGCCGCCGCTGATGC 3′ |
| EHMT1[B1-4]_EGFP (Reverse) | 5′ TAAGCAGGTACCTCGTGGCCTTTCTTGGCAGCCA 3′ |
| EHMT1[V09]_EGFP (Forward) | 5′ TCACTCAGATCT GCCACCATGGCTGCTGATGAAGGTTCCAC 3′ |
| EHMT1[V09]_EGFP (Reverse) | 5′ TAAGCAGGTACCTCTAGGGGGTCGGCGGCA 3′ |

For co-transfection of N and P, the 2 plasmids were used at a 1:1 ratio. For N, P, and L co-transfection as well as the mini-replicon system, the plasmids were used either at 3:6:2 or 1:1:0.5 ratio. Additionally, for mini-replicon, the SeV DI RNA was transfected using Lipofecta-mine 3000 12 h post transfection of N, P, and L.

## RNA extraction and qRT-PCR

Harvested cells were resuspended in Trizol for total RNA extraction by the conventional method using chloroform for phase separation and Isopropanol for precipitation of RNA. Reverse transcription of RNA to cDNA was performed using the PrimeScript RT Reagent kit (RR037A, Takara). Quantitative RT-PCR was then performed on the cDNA using TB Green Premix Ex Taq II (RR820A, Takara), following the instructions manual. Primer sequences used for qRT-PCR are provided in Table 2.

## Nuclear cytoplasmic fractionation and western blotting

The cells were harvested by trypsinization, and the pellet was washed twice with PBS. The pellet was then resuspended in 0.1% NP40 in PBS, supplemented with 1× protease inhibitor cocktail (PIC) and triturated about 10 times on ice. The samples were then incubated on ice for 1 min and centrifuged at 1,000g for 10 min at 4˚C. The resulting supernatant was the cytoplasmic fraction, which was aspirated into a fresh tube. The pellet was washed twice by resuspending in 0.1% NP-40 in PBS and centrifugation at 1,000g for 10 min at 4˚C. The resulting pellet was then resuspended in RIPA buffer + 1X PIC, incubated on ice for 1 h and vortexed at regular intervals. The samples were then centrifuged at 12,000g for 10 min at RT to eliminate the debris, the resulting supernatant was the nuclear fraction.

**Table 2. Primers used for qRT-PCR.**

| | |
|---|---|
| β-actin (Forward) | 5′ GCGTACAGGGATAGCACAGC 3′ |
| β-actin (Reverse) | 5′ GGATTCCTATGTGGGCGACGA 3′ |
| GAPDH (Forward) | 5′ GTCTCCTCTGACTTCAACAGCG 3′ |
| GAPDH (Reverse) | 5′ ACGACCCTGTTGCTGTAGCCAA 3′ |
| SeV gRNA (Forward) | 5′ ACTTAGGGTGAAAGCTCATCCA 3′ |
| SeV gRNA (Reverse) | 5′ ATCTTGATCCATGCGGGAAGT 3′ |
| SeV D1 RNA (Forward) | 5′ CACAACCGTGGTTGATGATGG 3′ |
| SeV DI RNA (Reverse) | 5′ ACCAGACAAGAGTTTAAGAG 3′ |
| IFN α (Forward) | 5′ TGAGACCCACAGCCTGGATA 3′ |
| IFN α (Reverse) | 5′ CTGGAGCCTTCTGGAACTGG 3′ |
| IFNβ (Forward) | 5′ AGCAGTTCCAGAAGGAGGAGGAC 3′ |
| IFNβ (Reverse) | 5′ TGATAGACATTAGCCAGGAGGTT 3′ |
| Ifit2 (Forward) | 5′ AGCAGCCTACGGCAACTAAA 3′ |
| Ifit2 (Reverse) | 5′ GCCTCGTTTTGCCCTTTGAG 3′ |
| ISG20 (Forward) | 5′ GGAGGGGATTGCTCCCTTGC 3′ |
| ISG20 (Reverse) | 5′ TTTCAGTGCCTGGAAGTCGT 3′ |
| IL8 (Forward) | 5′ AGACAGCAGAGCACACAAGC 3′ |
| IL8 (Reverse) | 5′ ATGGTTCCTTCCGGTGGT 3′ |
| IRF3 (Forward) | 5′ ACCAGCCGTGGACCAAGAG 3′ |
| IRF3 (Reverse) | 5′ TACCAAGGCCCTGAGGCAC 3′ |
| TNFα (Forward) | 5′ CCGTCTCCTACCAGACCAAG 3′ |
| TNFα (Reverse) | 5′ AGTCGGTCACCCTTCTCCAG 3′ |

Whole cell lysate was prepared by resuspending the cell pellet in RIPA buffer + 1X PIC, which was incubated on ice for 1 h and vortexed at regular intervals.

The lysates and fractions were estimated for protein concentration by Bradford Assay. Required amounts of the protein were reduced and denatured by mixing with 1X NuPAGE Sample Reducing Agent (Thermo Fisher Scientific) and IX NuPAGE LDS Sample Buffer and heating at 70°C for 10 min. The samples were then resolved on either 8% or 4% SDS-PAGE gel and immunoblotted with respective antibodies. The bands obtained by western blotting were quantified by ImageJ.

## Antibodies, inhibitors, and reagents

EHMT1 (NBP1-77400, Novus Biologicals, Rabbit polyclonal) was used for ICC, EHMT1 (A301-642A, Thermo Fisher Scientific, Rabbit polyclonal) was used for ICC, WB, IP. Sendai virus (ab33988, abcam, Chicken polyclonal), Sendai virus (PD029, MBL, Rabbit polyclonal), EHMT2 (NBP2-13948, Novus Biologicals, Rabbit polyclonal), Ezh2 (D2C9, Cell Signalling Technology, Rabbit monoclonal), Oct4 (9B7, MAI-104, Invitrogen, Mouse monoclonal), Hsp70 (W27, sc-24, Santa Cruz, mouse monoclonal), Gapdh (G9545, Sigma, Rabbit polyclonal), panH3 (ab1791, abcam, Rabbit polyclonal), H3K9me2 (ab32521, abcam, Rabbit monoclonal), LaminB1 (ab16048, abcam, Rabbit polyclonal), mCherry (NBP1-96752, Novus Biologicals, mouse monoclonal), Methylated e-N Lysine (ICP0501, Immunechem, Rabbit polyclonal), Dengue Virus NS1 protein antibody (GTX124280, GeneTex, Rabbit polyclonal), Chandipura Virus–L antibody was kindly provided by Dr. Dhrubajyoti Chattopadhayay (check with Nishi how to acknowledge this). Normal Rabbit IgG (12–370, Sigma-aldrich, Rabbit polyclonal), Normal Mouse IgG (12–371, Sigma-aldrich, Mouse polyclonal).

For ICC, Alexa Fluor conjugated secondary antibodies from Invitrogen were used: Goat anti-Rabbit 633 (A21071), Goat anti-Rabbit 568 (A11011), Goat anti-chicken 568 (A11041), Goat anti-mouse 488 (A11001), Goat anti-Rabbit 488 (A11008), Goat anti-mouse 568 (A11004). HRP-conjugated secondary antibodies were used for western blotting: Goat anti-Rabbit (1706515, Biorad), Goat anti-mouse (1706516, Biorad). BIX-01294 (B9311, Sigma), UNC0642 (SML1037, Sigma). Dynabeads Protein A (10001D, Thermo Fisher Scientific), Dynabeads Protein G (10003D, Thermo Fisher Scientific).

## Cell infection, treatment, and transfection

Cells were infected with WT SeV, Cytotune SeV, or EmGFP SeV at a multiplicity of infection of 2 in culture media. DenV and ChpV were also infected at an MOI of 2. Cells were treated with 3 μm of BIX-01294, 3 μm of UNC-0642, or 10 μm of BX795 which was replenished every 24 h. Plasmids were transfected in HEK by using Xfect Transfection Reagent (631318, Takara) by following the user manual. SeV DI RNA was transfected using Lipofectamine 3000 Transfection reagent (L3000001) by following the user manual.

## T7 endonuclease assay

Genomic DNA was isolated and 550 bp region of the exon 3 was amplified, with about 250 bases flanking on either side of the gRNA targeted region.

## Mini replicon system

The SeV DI RNA was designed as described by Calain and colleagues [46,47]. The full-length Sendai viral genome is about 15 Kb long; the Defective-Interfering DI-H genome about 1/10th the length of the full-length RNA, which is about 1,400 nucleotides. The DI genome contained

the 5' trailer sequence, along with a stretch of the L-gene, and has a copy-back termini with 110 nucleotides of the 3' terminus, complementary to the 5' end. This sequence forms a ssRNA loop with dsRNA panhandle-like structures and possess replicative advantage over WT full-length viral genome.

In this study, we created a minireplicon system for SeV by co-transfecting HEK with the N, P, and L plasmids in a 3:6:2 or 1:1:0.5 ratio, and 12 h post transfection of the plasmids, the SeV DI RNA was transfected. We chose 12 h as a time point to transfect DI RNA because at this point, viral proteins would be expressed in cells to encapsidate the RNA upon entry. At this point, viral IBs also were not formed, so this enabled us to treat the cells with EHMT inhibitors while simultaneously transfecting with the RNA, which allowed us to study IB formation.

## Immunoprecipitation and RIP

Dynabeads protein A or G were washed twice with 0.1% BSA in 1 M potassium phosphate buffer. To this, 2 μg of the respective antibody diluted in 0.5 M potassium phosphate buffer was added and incubated on a rotor for 2 h at RT. The unbound antibody was then washed with potassium phosphate buffer and the protein lysate, or the cytoplasmic fraction was added and incubated ON at 4˚C. (For immunoprecipitation, the cytoplasmic fraction was mixed 1:1 with RIPA buffer.) On the following day, the unbound fraction was removed by washing with IP-100 and 1X Flag buffer. The immunoprecipitated protein was then eluted by incubating in 2× reducing agent and protein loading dye at 70˚C for 10 min.

## RNA immunoprecipitation

The cells were washed with PBS in the culture dish and UV-crosslinked at 450 mJ/cm$^2$ before proceeding with Nuclear cytoplasmic fractionation or cell lysis. About 750 to 1,000 μg of protein was added to the antibody bound dyna beads and incubated ON at 4˚C. On the following day, the unbound fraction was removed by washing with IP-100 or 1× Flag buffer. Immuno-precipitated RNA was eluted from the beads by treating with 5 mg/ml ProteinaseK for 15 min at 37˚C; the sample was then incubated with trizol for RNA extraction by the chloroform/iso-propanol method.

## Haemolysis assay

Whole human blood was collected in tubes treated with an anticoagulant. Required volume of whole blood was diluted 50 times with 1× PBS and centrifuged at 3,000 rpm for 10 min at 4˚C thrice. The pellet containing RBCs was diluted to obtain 1% RBCs solution. To this, culture supernatant from infected cells ± inhibitors with appropriate controls was added and incubated for 45 min in ice for attachment of viral particles to the surface of RBCs; this was followed by centrifugation at 3,000 rpm for 10 min at 4˚C. The pellet was resuspended in 1× PBS and incubated at 37˚C for 30 min in a shaking incubator, followed by centrifugation at 3,000 rpm for 10 min at 4˚C. The supernatant was collected in a 96-well plate and absorbance was measured at 540 nm.

## Statistical analysis

Graphpad Prism 8 was used to perform all the statistical analyses. For analysis of 2 groups, paired or unpaired $t$ test was used according to the experiment. For comparison between 3 or more groups, one-way ANOVA or Brown–Forsythe and Welch ANOVA was used. Details of $N$ (replicate size), $n$ (sample number), and the tests applied have been mentioned in the respective figure legends; $p$-value: 0.1234 (ns), 0.0332 (*), 0.0021 (**), 0.0002 (***), <0.0001 (****).

## Supporting information

**S1 Fig. Transduction of Sendai virus encoding for OSKM in fibroblasts resulted in formation of EHMT1 condensates in the cytoplasm.** Confocal microscopic images of (A) various stages of fibroblasts undergoing reprogramming induced by the ectopic expression of OSKM delivered via SeV, immunolabelled with EHMT1 (green). (B) Fibroblasts transduced with OSKM SeV, immunolabelled with EHMT1 (green), at indicated time points. (C) Fibroblasts transduced with OSKM via SeV, immunolabelled for EHMT1 (red) and Oct4 (green). (D) Fibroblasts transfected with episomal plasmids expressing OSKM, immunolabelled with EHMT1 (red) and Oct4 (green). (E) Uninfected HEK immunolabelled with EHMT1 (red), (F) uninfected BEAS-2B immunolabelled with EHMT1 (red). (G) Mouse embryonic fibroblasts (MEFs) infected with SeV, immunolabelled with EHMT1 (red). (H, I) BEAS-2B infected with EmGFP SeV (green) immunolabelled with (H) EHMT2 (red) and (I) Ezh2 (red). Composite of images are with DAPI (blue) stained nuclei. Scale bar, 40 μm. Raw confocal microscopic images are deposited on BioImage Archive (Accession id: S-BIAD1362).
(TIF)

**S2 Fig. Characterisation of SeV replication organelles.** Confocal microscopic images of (A) BEAS-2B infected with EmGFP SeV (green) immunolabelled with EHMT1 (red) and SeV (grey). HEK transfected with (B) mCh_N (red) and (D) piRFP_P (yellow), (C) HEK transfected with mCh_N (red) and immunolabelled with EHMT1 (yellow), (D) HEK transfected with piRFP_P (yellow) and immunolabelled with EHMT1 (red). (F) HEK transfected with EGFP_L (green). (G) Graph representing the overall area of all IBs formed in cells across infection and co-transfection conditions. (*n* = 3 replicates, unpaired *t* test with Welch's correction between the indicated groups, *p*-value: 0.1234 (ns), 0.0332 (*), 0.0002 (***)) (H) Fibroblast uninfected and (I) infected with SeV co-immunolabelled with Hsp70 (red) and EHMT1 (green). (J) HEKs co-transfected with mCh_N (red) and piRFP_P (yellow) in a 1:1 ratio, immunolabelled with Hsp70 (green). Composite of all images are with DAPI (blue) stained nuclei. Scale bar, 40 μm. Source data are provided as S1 Data. Raw confocal microscopic images are deposited on BioImage Archive (Accession id: S-BIAD1362).
(TIF)

**S3 Fig. Distinct cytoplasmic form of EHMT1 localizes with SeV IBs.** (A) Whole cell lysate of HEKs transfected with mCh_C1 or mCh_EH1_FL, western blotted and probed with mCherry. Confocal microscopic images of HEK transfected with (B) mCh_EH1_FL (red), (C) mCh_EH1_FL (red), infected with EmGFP SeV (green) and immunolabelled with SeV (yellow). Nuclei stained blue with DAPI. (D) Graph representing relative protein levels of EHMT1 quantified from western blots of nuclear and cytoplasmic fractions at 24 h p.i. EHMT1 bands from SeV infected lanes were normalised against uninfected lanes (*n* $\geq$ 3 replicates, one-way ANOVA, *p*-value: 0.0332 (*), 0.0021 (**)). (E, G) HEK were transfected with either mCh_EH1_FL, EHMT1$^{B1-3}$_Egfp, EHMT1$^{B1-4}$_Egfp, or EHMT1$^{V09}$_Egfp. Table representing (E) the percentage of transfected cells demonstrating nucleo-cytoplasmic signal, (G) co-transfected with mCh_N and piRFP_P; table representing % triple transfected cells with colocalization of various forms of EHMT1 into IBs. (F) Confocal microscopic images of fibroblasts immunolabelled with EHMT1 (red) acquired by adjusting the threshold corrected for background with a secondary control. (H) Schematic representation of EHMT1 truncated sequences cloned into EGFP reporter plasmid. (I) Confocal microscopic images of HEK transfected individually with N300_EGFP, Delta EHMT1_EGFP and Ank+SET_EGFP. (J) Confocal microscopic images of HEK transfected with N300-GFP, Delta EHMT1_EGFP and Ank+SET_EGFP, co-transfected with mCh_N + piRFP_P, composite images are with DAPI (blue)

stained nuclei. Scale bar, 40 μm. Source data are provided as S1 Data. Raw confocal microscopic images are deposited on BioImage Archive (Accession id: S-BIAD1362).
(TIF)

**S4 Fig. EHMT1$^{N/C}$ influences the formation of large IBs, thereby regulating viral replication.** (A) Confocal microscopic composite images of EmGFP SeV (green) infected BEAS-2B cells, simultaneously treated with DMSO, 3 μm BIX or 3 μm UNC, immunolabelled with SeV ab (red), marking the IBs. (B) Graph plotted for the relative expression of SeV gRNA assessed by qRT-PCR with the Ct values normalised against GAPDH ($n$ = 4 replicates, one-way ANOVA, **, $p < 0.005$). (C) Confocal microscopic composite images of HEK transfected with mCh_N (red) and piRFP_P (yellow), treated with DMSO, 3 μm BIX or 3 μm UNC. (D) Confocal microscopic images of HEK co-transfected with EGFP_L (green), mCh_N (red), and piRFP_P (grey), treated with DMSO, 3 μm BIX or 3 μm UNC. Composite images are with DAPI (blue) stained nuclei. Scale bar, 40 μm. (E) Graph representing the relative expression of SeV DI RNA at indicated time points post transfection, as assessed by qRT-PCR with Ct values normalised against GAPDH. ($n$ = 4 replicates, one-way ANOVA, $p < 0.005$ (**), $p < 0.05$ (*)) Source data are provided as S1 Data. Raw confocal microscopic images are deposited on BioImage Archive (Accession id: S-BIAD1362).
(TIF)

**S5 Fig. Depletion of EHMT1$^{N/C}$ and inhibition of IFN signalling.** (A) Schematic and sequence of EH_132 guide RNA targeting exon 3 of EHMT1 via CRISPR/Cas9 method of gene editing. (B) Agarose gel electrophoresis of the products from T7 endonuclease assay demonstrating gene editing as seen by digestion into 3 products in EH_132 mutated sample. (C) Western blotting of the SpCas9 control and EH_132 lysates immunoblotted with EHMT1, GAPDH, H3K9me2, and panH3. (D) Western blotting of the whole cell, nuclear and cytoplasmic fractions of single cell clones of EH_132, probed with EHMT1, LaminB1, and GAPDH. (E, F) Graphs representing relative expression of Type 1 IFN and stimulated genes over the time course of infection with WT SeV, as assessed by qRT-PCR. (G–J) HEK were transfected with SpCas9 or E132 (EH_132), infected with WT SeV and simultaneously treated with BX795; qRT-PCR was performed to assess the relative expression of (G) IFNα, (H) IFNβ, (I) ISG20, and (J) Ifit2 across the indicated conditions. ($n$ = 4 replicates, one-way ANOVA, $p < 0.05$ (*), $p < 0.005$ (**), 0.0002 (***), <0.0001 (****)) Source data are provided as S1 Data.
(TIF)

**S6 Fig. Cytoplasmic EHMT1 and EHMT2 associate with the SeV nucleoprotein.** (A) EHMT1 IP from the cytoplasmic fraction of SeV infected or uninfected cells, elute western blotted and probed with EHMT1 and SeV. Dotted line in black indicates the nucleo-cytoplasmic form of EHMT1. (B) EHMT1 IP from the cytoplasmic fraction of cells transfected with mCh_N, infected with SeV, elute western blotted and probed with mCherry. (C) mCherry IP from whole cell lysate of HEK transfected with mCh_EH1_FL and infected with EmGFP SeV; the elute was western blotted and probed with mCherry and SeV, depicting no association between mCh_EHMT1_FL and SeV proteins. (D) GFP IP from cells triple transfected with EHMT1$^{B1-4}$_EGFP, mCh_N and piRFP_P; elute was western blotted and probed with GFP and mCherry. (E) EHMT1 IP from the cytoplasmic fractions of uninfected or SeV infected cells; elute western blotted and probed with EHMT2, depicting an interaction between EHMT1 and EHMT2. (F) EHMT2 IP from the cytoplasmic fraction of SeV infected and uninfected cells, elute western blotted and probed with SeV and EHMT2. (G, H) Graphs representing quantification of the (G) SeV nucleoprotein and (H) meK levels from western blots (Fig 6I). For (G), protein levels were normalised against GAPDH and for (H), protein levels

were normalised against the respective nucleoprotein bands. ($n$ = 4 replicates, one-way ANOVA, **, $p < 0.005$, 0.0002 (***), <0.0001 (****)) (I–K) HEK were co-transfected with mCh_N and piRFP_P, EHMT1$^{N/C}$ was immunoprecipitated from the cytoplasmic fraction 24 h post transfection. IP elute was western blotted and probed with (I) EHMT1, (J) mCherry, and (K) meK. Source data are provided as S1 Data.
(TIF)

**S7 Fig. Characterisation of IB formation in infected cells.** (A) Graphs representing the mean number of IBs (y-axis) in each subpopulation for WT SeV (pink line) infected cells. Data are ±SEM. ($n > 25$ cells, Brown–Forsythe and Welch ANOVA tests) $p$-value: 0.1234 (ns), 0.0332 (*), 0.0021 (**), 0.0002 (***), <0.0001 (****). (B) Confocal microscopic images of BEAS-2B cells infected with EmGFP SeV (green) immunolabelled with EHMT1 (red) and SeV (grey) at indicated time points postinfection, composite images of all channels are with DAPI (blue) stained nuclei. Scale bar, 40μm. (C) Graphs representing the mean number of IBs (y-axis) in each subpopulation for EmGFP SeV (green line) infected cells. Data are ±SEM. ($n > 25$ cells, Brown–Forsythe and Welch ANOVA tests) $p$-value: 0.1234 (ns), 0.0332 (*), 0.0021 (**), 0.0002 (***), <0.0001 (****). Source data are provided as S1 Data. Raw confocal microscopic images are deposited on BioImage Archive (Accession id: S-BIAD1362).
(TIF)

**S8 Fig. Enzymatic activity of EHMT1$^{N/C}$ facilitates the formation of large IBs.** (A–N) Graphs representing the number and area of IBs as quantified from confocal microscopic images using the ImageJ analysis tool. (A–C) BEAS-2B cells were infected with WT SeV and treated with 3 μm of BIX/UNC simultaneously. The cells were fixed and immunolabelled with SeV antibody 16 h p.i. Graphs representing number of IBs in (A) 0.1–1 μm$^2$ category, (B) 1–10 μm$^2$ category, and (C) >10 μm$^2$ category. ($N = 3$ replicates, $n > 100$ cells, ordinary one-way ANOVA, $p$-value: 0.1234 (ns), 0.0332 (*), 0.0002 (***), <0.0001 (****)) (D) Graph representing the total number of IBs of various subpopulations in N+P co-transfected cells, treated with 3 μm of BIX/UNC. ($N = 5$ replicates, $n > 300$ cells, ordinary one-way ANOVA, 0.1234 (ns)) (E–G) HEK were co-transfected with mCh_N, piRFP_P, and EGFP_L, followed by transfection with SeV DI RNA 12 h post transfection of the plasmids, either untreated or treated with 3 μm of BIX/UNC. Graphs representing number of IBs in (E) 0.1–1 μm$^2$ category, (F) 1–10 μm$^2$ category, and (G) >10 μm$^2$ category. ($N = 2$ replicates, $n > 200$ cells, ordinary one-way ANOVA, 0.1234 (ns), <0.0001 (****)). (H–J) HEK were transfected with empty SpCas9 or EH_132 plasmid to deplete the levels of EHMT1 and infected with WT SeV, cells were immunolabelled with SeV 16 h p.i. Graphs representing number of IBs in (H) 0.1–1 μm$^2$ category, (I) 1–10 μm$^2$ category, (J) >10 μm$^2$ category. ($N = 3$ replicates, $n > 100$ cells, unpaired $t$ test) $p$-value: 0.1234 (ns), 0.0021 (**), <0.0001 (****)). (K, L) HEK293 were transfected with SpCas9 empty plasmid or EH_132, and mCh_EH1_FL (mEL); the cells were then infected with WT SeV and immunolabelled for SeV 16 h p.i. (K) Graph representing the total number of IBs and (L) area of IBs >10 μm$^2$. ($N = 2$ replicates, $n > 100$ cells, unpaired $t$ test) $p$-value: 0.1234 (ns), <0.0001 (****)) (M) HEKs were transfected with SpCas9 or EH_132, infected with WT SeV and simultaneously treated with BX795. Graph representing the number of IBs >100 μm$^2$. ($N = 3$ replicates, $n > 100$ cells, unpaired $t$ test) $p$-value: 0.0021 (**)). Data are ±SEM. Source data are provided as S1 Data.
(TIF)

**S1 Data. Source file containing numerical values of all the graphs and tables.**
(XLSX)

**S1 Raw Images. PDF file containing uncropped, unedited western blots from all the figures.**
(PDF)

**S1 Movie. Live cell microscopic imaging of HEK transfected with mCh_N (red) and infected with EmGFP SeV (green).**
(AVI)

## Acknowledgments

We thank Prof. Colin Jamora and Dr. Dasaradhi Palakodeti for their critical and constructive feedback on the manuscript. We thank Dr. Sheetal Gandotra for the Volocity Image Analysis software and helpful discussions. The Central Imaging and Flow Cytometry Facility (CIFF) at InStem and NCBS, Confocal microscopy facility at CSIR-IGIB, Confocal microscopy facility at Shiv Nadar University supported the Confocal microscopic imaging.

## Author Contributions

**Conceptualization:** Kriti Kestur Biligiri, Shravanti Rampalli.

**Data curation:** Kriti Kestur Biligiri.

**Formal analysis:** Kriti Kestur Biligiri.

**Funding acquisition:** Shravanti Rampalli.

**Methodology:** Kriti Kestur Biligiri, Shravanti Rampalli.

**Project administration:** Kriti Kestur Biligiri, Shravanti Rampalli.

**Resources:** Nishi Raj Sharma, Abhishek Mohanty, Debi Prasad Sarkar, Praveen Kumar Vemula.

**Supervision:** Debi Prasad Sarkar, Shravanti Rampalli.

**Validation:** Kriti Kestur Biligiri.

**Visualization:** Kriti Kestur Biligiri.

**Writing – original draft:** Kriti Kestur Biligiri, Shravanti Rampalli.

**Writing – review & editing:** Kriti Kestur Biligiri, Nishi Raj Sharma, Debi Prasad Sarkar, Praveen Kumar Vemula, Shravanti Rampalli.

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
