## [Editor Report · Decision Letter 0]

6 Dec 2023

Dear Dr Rampalli, 

Thank you for submitting your manuscript entitled "Novel nucleo-cytoplamic form of Euchromatic Histone Methyltransferase1 (EHMT1N/C) methylates RNA viral proteins involved in inclusion body formation to facilitate viral replication" for consideration as a Research Article by PLOS Biology.

Your manuscript has now been evaluated by the PLOS Biology editorial staff as well as by an academic editor with relevant expertise and I am writing to let you know that we would like to send your submission out for external peer review.

Once your full submission is complete, your paper will undergo a series of checks in preparation for peer review. After your manuscript has passed the checks it will be sent out for review. To provide the metadata for your submission, please Login to Editorial Manager (https://www.editorialmanager.com/pbiology) within two working days, i.e. by Dec 08 2023 11:59PM.

Kind regards,

Nonia

Nonia Pariente, PhD, 

Editor-in-Chief

PLOS Biology

npariente@plos.org

---

## [Decision Letter · Decision Letter 1]

15 Jan 2024

Dear Dr Rampalli,

Thank you for your patience while your manuscript "Novel nucleo-cytoplamic form of Euchromatic Histone Methyltransferase1 (EHMT1N/C) methylates RNA viral proteins involved in inclusion body formation to facilitate viral replication" was peer-reviewed at PLOS Biology. Your manuscript has been evaluated by the PLOS Biology editors, an Academic Editor with relevant expertise, and by several independent reviewers.

As you will see in the reviewer reports, although the reviewers acknowledge the potential interest in your findings, they have also raised a substantial number of crucial concerns. Based on their specific comments and following discussion with the Academic Editor, it is clear that a substantial amount of work would be required to meet the criteria for publication in PLOS Biology. Given our and the reviewer interest in your study, we would be open to inviting a comprehensive revision of the work; however, this would need to thoroughly address all the reviewers' comments in full, specially since reviewer #2 suggests rejection. A successful revision will need to resolve issues related to the experimental rigor necessary to support the conclusions, such as providing additional results supporting the recruitment of a specific form of EHMT1 to inclusion bodies, elucidating the correlation between the size of inclusion bodies and the level of viral replication, and providing a rationale behind the reduction in the size of inclusion bodies and SeV genome. Furthermore, the revision must clarify the data used and its interpretation, improving the manuscript's structure for enhanced readability, and addressing other concerns regarding wording and presentation.

Given the extent of revision that would be needed, we cannot make a decision about publication until we have seen the revised manuscript and your response to the reviewers' comments. Your revised manuscript would need to be seen by the reviewers again, but please note that we would not engage them unless their main concerns have been addressed.

We appreciate that these requests represent a great deal of extra work, and we are willing to relax our standard revision time to allow you 6 months to revise your study. Please email us (plosbiology@plos.org) if you have any questions or concerns, or envision needing a (short) extension.

**IMPORTANT - SUBMITTING YOUR REVISION**

*Resubmission Checklist*

*Published Peer Review*

*PLOS Data Policy*

*Blot and Gel Data Policy*

Sincerely,

Melissa

Melissa Vazquez Hernandez, Ph.D.

Associate Editor

PLOS Biology

REVIEWERS' COMMENTS:

Reviewer #1: 

In this manuscript, Biligiri et al. reported that a novel nucleo-cytoplamic form o f EHMT1, which is a histone methyltransferase, can interact with N and P protein of the Sendai virus and promote inclusion body formation upon virus infection. They explored the underlying mechanism and suggested a new mechanism by which cytoplasmic EHMT1 acts as proviral host factor that regulates host-pathogen interface to discern RNA viral pathogenesis through phase separation (or condensation) of the inclusion bodies.

This new discovery is interesting, and provided the community to appreciate the association between histone methyltransferase and virus inclusion bodies. There are some concerns needed to be addressed so their conclusion can be more solid:

The authors could try 1-2 more virus to infect the cells and see whether EHMT1 still can form condensates.

Dose EHMT2 also can form condensates upon virus infection?

Dose EHMT1 or the N, P proteins has IDR domain (intrinsically disordered regions)? And What is their function?

Figure 3, The authors used IP-EHMT1 to pull down the N and P proteins; could the use IP-N or IP-P to pull down the EHMT1 protein?

Please mark the protein size (i.e. kilodalton) in the Western blots.

Why the full-length vector of EHMT1 can not enter the cytoplasmic? Can the authors give any explanations?

The reviewer is wondering that Delta EHMT1 is indeed existed in the real-world? Can the authors give and evidences? What is ratio of Delta EHMT1 vs full-length EHMT1? Who is responsible for the making of Delta EHMT1?

Did the two inhibitors specifically targeting EHMT1, can the authors give more evidences?

In respect to RNA virus replication, the authors only used the gRNA as the readout. They should give more readouts, since virus replication is the most important and the only function assay in this study. 

For Figure 5 and Figure 7, the authors could provide an excel file to include their original date obtained from their microscope observation on inclusion bodies numbers and sizes. 

For Figure 6, the authors should perform in vitro experiment to confirm that EHMT1 can demethylate N protein. 

Can the authors provide any in vivo data to support their discovery? 

All of the virus infection experiment is only last 1-3 days, the reviewer is wondering can we see the same phenomena in a stable infection situation?

Reviewer #2:

In this article, the authors describe the recruitment of an EHMT1 methyltransferase to Sendai virus IBs. This recruitment increases as the virus cycle progresses and coincides with an increase in the size of IBs. The authors show that EHMT1 is a proviral factor. EHMT1 interacts with the RNPs of the virus. It would methylate the N and P of Sendai virus. Unfortunately, the experiments need to be completed to support the conclusions presented by the authors. In addition, the manuscript needs extensive restructuring to make it more readable.

Firstly, a lack of rigour in the experiments and/or in the presentation of the results limits the interpretation of the results and the soundness of the conclusions. In particular data presented are not strong enough to support results 3, 4 and 6.

Result 1 (lines 104-181): EHMT1 is recruited to Sendai IBs and pseudo-IBs generated by expression of N and P

Infection of different cell types with Sendai virus (wild-type and EGFP) showed translocation of EHMT1 to IBs. The same was observed in pseudo-IBs formed with N and P alone. The first 2 paragraphs are redundant and the figures are not very informative. The uninfected panel is missing from Figures 1 and 2 for full interpretation. 

The figures should be combined into a complete and coherent main figure and one or two sup-figures.

Result 2 (lines 182-251): A distinct form of EHMT1 associates with IBs

The authors show a band of lower MW than EHMT1 recognised by the antibody against EHMT1. Observation of the disappearance of this band under conditions of EHMT1 under-expression would complete the demonstration. Note that this decrease is observed in Sup Fig 4D, but on a less resolving gel.

To show that this particular form of EHMT1 is recruited to IBs, the authors rely on 2 results:

1) Immunocapture of N and P by anti-EHMT1 in the cytosolic extracts, where only this form should be present. For a more robust demonstration, parallel migration of nuclear fractions (on a 4% gel) is required.

2) The authors used transient transfections with vectors expressing either the full protein or truncated forms. The localisation of these fluorescent proteins in uninfected and infected cells was analysed by immunofluorescence. The results are presented in 3 different figures, making them difficult to read. The infection state is not tested for all mutants and only one cell is shown. There are no results showing that the smaller MW form is close to the delta EHMT1 form constructed by the authors. Although these results support the hypothesis put forward, they do not allow us to conclude on the nature of the N/C form of EHMT1.

Result 3 (lines 182-251): Interaction between EHMT1 and viral and cellular partners. 

It would be preferable to separate this result from the identification of the N/C form to clarify the message by first presenting these interaction data on total extracts. RNA capture data should also be presented here.

Note that there is always non-specific precipitation of N, which makes the comment about N and P capture levels (line 213) questionable. 

To further investigate the nature of the captured complex, the authors express mCh-N alone. The results are shown in Figure 3F. The input for the IgG control, which is necessary for the interpretation of the experiment as N tends to be immunoprecipitated specifically, is missing. 

The reason for transfecting mCh-N with an infection is unclear. The combination of infection and transfection is highly inefficient (the 2 events inhibit each other). Transfected infected cells can be detected and analysed by immunofluorescence, but this is difficult to interpret by immunoprecipitation. 

To determine the nature of the EHMT1 viral interactor, immunoprecipitation must be performed with cells transfected to express N alone, P alone and P + N.

Result 4 (lines 252-303): Inhibition/KO of EHMT1 limits viral replication

The qPCR protocol on which this conclusion is based needs to be detailed. It is not specified which primer is used for RT, how the authors ensure that the PCR is strand specific (or would it be better to talk about total viral RNA), whether the points are made in duplicate or triplicate, and whether there is any standardisation to cellular targets. 

A factor of 2 corresponds to a Ct difference that is very small and difficult to interpret. As Sendai virus is easy to titrate, at least some of these results should be confirmed by plaque assay titration.

Result 5: EHMTs methylate N and P

It would be interesting to see if N and P are methylated when transiently co-expressed in the absence of infection. This would confirm the link with the recruitment of EHMT1 to IBs.

Result 6: EHMTs play a role in IB growth

Results paragraphs lines 304-308 and lines 448-501 establish the link between EHMT1 recruitment, IB size increase, replication and methylation of viral proteins. 

The authors note a correlation between the size of IBs and the level of viral replication, but the results presented are insufficient to draw any conclusions. 1) They base their conclusions on the level of replication assessed by quantification of viral RNA using a non-detailed protocol. "Normalised Ct" should be defined. The normalised Ct results for the control (uninfected) are surprising (control Ct 10) and very different between the wild-type virus and the GFP virus (10 versus 25), and the T0, which is essential for interpreting the kinetics, is missing. It is necessary to detail the protocol (and explain the results if necessary), quantify the RNA at T0 and perform titrations in the lysis range to quantify the viral replication kinetics. 2) For the wild-type virus, there is no difference in "Ct" between 12, 24 and 48 hours. Note that the GFP virus behaved very differently (see specific comment on this point).

The authors observed a reduction in the size of IBs when EHMT1 was inhibited, but the results did not show that this reduction in size was the cause of the reduction in viral replication. The reduction in the amount of viral proteins alone could explain this phenomenon. The fact that EHMT1 inhibitors had no effect on the pseudo-IBs formed when N and P were expressed alone could be interpreted in the same way as the authors (in favour of an effect on large IBs alone) or, on the contrary, as an argument in favour of an effect on viral synthesis. In conclusion, it is necessary to use a mini-replicon system in which the amount of viral proteins does not depend on viral synthesis. 

Point 7: Problem with the Sendai GFP virus: This virus behaves differently from the wild-type virus: very different EHMT1 relocalisation profiles (Fig. sup1 and Fig. sup2F), very different evolution of the number and size of IBs (Fig. 5 C, D and sup Fig. 5) and different replication kinetics (Fig. 5E and F). When showing experiments based on this virus, it is necessary to indicate the conditions under which the virus was amplified, to indicate the genome structure of this virus and to compare its replication kinetics with those of the wild-type virus. 

Additional point: For all conclusions based on images without quantification, authors should indicate the number of experiments performed and the number of images taken (or cells observed).

Some minor comments

Sup Fig3D: Explain the method used. In the uninfected case, there is as much EHMT1 in the cytoplasm and more in the nucleus than in the whole cell? on the contrary, in the infected case, there is more in the 2 compartments ?

Lines 207-209: The authors conclude that there is a reduction in EHMT1 levels in infected cells. The results do not support this conclusion. The levels appear identical in Figure 3C, Figure Sup3D uses total extract as the standard for each condition, which does not allow comparison of levels between infected and uninfected cells.

Figure Sup3F If N means nuclear extract, why is EHMT2 abundant when it should be cytoplasmic?

RNA viral protein title: unclear concept. Are we talking about RNP?

For significance levels, do not give values, leave <0.05; <0.01; <0.001.

Reviewer #3: 

This manuscript shows that a cellular methyltransferase, EHMT1, can be found in the cytoplasm and localizes into SeV inclusion bodies (IBs), interacting with the virus P and N proteins. Inhibition of methyltransferase activity or knockout of EHMT1 causes a reduction of SeV IBs size and levels of SeV genomic RNA, suggesting a pro-viral role for this cellular protein. However, because EHMT1 has roles in preventing expression of the antiviral interferons, this reviewer is not convinced that the reduction of the IB size and SeV genome is not at least in part due to lack of IFN repression. This major concern can be resolved by measuring IFN transcripts in controls and KO cells. If IFN is expressed to higher levels, then IFN can be blocked (using KO cells for IFN signaling elements, or neutralizing antibodies) and test for the impact of EHMT1 methyltransferase in virus replication in the absence of this confounding factor. Another major solvable concern is the overstatement and generalization of conclusions in figures that clearly show differences in only one of the conditions tested. Examples of these below.

Additional comments:

Wording needs to be revise for clarity and simplicity. For example, in the abstract the authors conclude: "EHMT1 acts as proviral host factor that regulates host-pathogen interface to discern RNA viral pathogenesis". What does this means? This sentences makes no sense to me. 

Figure S2, D: Transfection of piRFP_P appears to cause aggregation of EHMT1 and these aggregates look very different from the ones observed during infection. However, these aggregates do not colocalize with P. What is the explanation for this observation?

Lines 162-167 and Fig S2, E, F, G: Unclear what the relevance of staining for Hsp70 and how it fits into the EHMT1 story. A better explanation and rationale for using these data should be provided for clarity. 

Figure 3A: Although there is clearly a band for EHMT1 in the cytoplasmic fraction, it is unclear why this signal is not observed in the images of the uninfected population of cells in Fig 1 and 2, especially considering there doesn't seem to be an upregulation of this protein when comparing uninfected and infected cells. According to the methods, the same antibody was used for both techniques. A note or an explanation for this should be provided. A possibility is that this has to do with the thresholds that are being set for the imaging where in some instances are removing signal that is not background. 

Figure 3F: Please explain why does mCherry tagged NP show two bands? And why are these bands a different size when pulling down EHMT1?

Fig 3G: what is the bottom band that appear specifically in the IP conditions? 

Line 262 and Figure 4C: generalizes an observation that is only found in relation to one of the drugs. Only UNC drug show differences while the BIX inhibitor shows no significant differences with the DMSO treated. This should be clearly stated and if possible, an explanation to these differences should also be provided.

Figure 4K: Although complementation of the nuclear version of the EHMT1 shows no rescue of the phenotype, it does not directly demonstrate that the host response is not involved in the phenotype observed. It is assuming that both the nuclear and the nucleo-cytoplasmic have equal roles in modulating antiviral responses, which is not known given that this article just discovered the existence of this alternate version of the protein, which according to Fig 3B, also localizes in the nucleus. A more simple and direct way to demonstrate that the antiviral responses regulated by EHMT1 are not orchestrating these differences in phenotype is a qPCR of IFN expression in the control vs the knockout cells. Unchanged levels of IFN in these cell lines will directly confirm the role of EHMT1 in IB size and replication is independent on its role regulating IFN. 

Figure 5C: Please explain how the boundaries of the cell were defined to calculate number of IBs per cell.

Line 326-328 and Figure 5C: generalizes an observation that is only shown in the EmGFP infection. There is no change between the first 36 hrs in the WT infection. This should be noted and an explanation to these differences should be provided. 

Line 246 to 248 and Figure 5E and 5F: Unclear where the peak of replication is demonstrated. The representation of the statistics is confusing. Comparison should be made in between time points to demonstrate a "peak" of infection. There is no significant difference between times 12-48 hpi in figure 5E, demonstrating that there is not "peak" at 24 hrs. 

Line 399: Should clarify that the methylation observed for the P protein is extremely minimal since the band is so faint, almost invisible. 

Line 77: missing a "the" before cytoplasmic

Line 136: Should say "single-strand negative-sense RNA viruses"

Line 158: ICC is never defined

Figures 4A, 4D and 4G: These diagrams are unnecessary. The approach is straight forward and these diagrams pollute the figure. 

Line 259: Unsure what "compact" means in relation to the comparison of the IBs in the different conditions. 

Line 298: missing a "be" after the "not"

Figure S5A: Graphs missing legend 

Figure 7: Hard to differentiate the normal NP with the methylated version. Perhaps another color and a bigger doodle for the methylation would make it easier to see.

---

## [Decision Letter · Decision Letter 2]

6 Sep 2024

Dear Dr Rampalli,

Thank you for your patience while we considered your revised manuscript "Novel nucleo-cytoplamic form of Euchromatic Histone Methyltransferase1 (EHMT1N/C) methylates RNA viral proteins involved in inclusion body formation to facilitate viral replication" for publication as a Research Article at PLOS Biology. This revised version of your manuscript has been evaluated by the PLOS Biology editors, the Academic Editor and two of the original reviewers, one of them being Jian Ma. I would like to apologize for the delay, but reviewer recruitment has particularly challenging this time.

Based on the reviews and on our Academic Editor's assessment of your revision, we are likely to accept this manuscript for publication, provided you satisfactorily address the remaining editorial points. Please make sure to address the following data and other policy-related requests.

a) We routinely suggest changes to titles to ensure maximum accessibility for a broad, non-specialist readership, and to ensure they reflect the contents of the paper. In this case, we would suggest a minor edit to the title, as follows. Please ensure you change both the manuscript file and the online submission system, as they need to match for final acceptance:

"A cytoplasmic form of EHMT1N methylates viral proteins to enable inclusion body maturation and efficient viral replication"

b) In the abstract, please correct ""In this study, we discovered distinct nucleo-cytoplasmic formS of Euchromatic Histone

Methyltransferase1(EHMT1N/C 46 ), a PKMT, that phase separates into viral inclusion bodies (IBs) upon

cytoplasmic RNA-virus infection (Sendai Virus)". The "s" in formS, should not be there. 

c) We believe that some figures shown as response to reviewers are a more complete version of the figures. Please use the complete version shown in Reference Figure 1 for Fig 9. And please add the DAPI controls shown in reference figure 2A and 14 to figures S1G and S2DE, respectively. Please also add the reference Figure 5 as a supplementary material. 

d) The Ethics statement needs to be the first subheading in the Methods sections the Material & Methods section. You currently have it in the subsection for the haemolysis assay; please move it to the beginning of Materials and Methods in its own subsection.

Please supply the numerical values either in the a supplementary file or as a permanent DOI’d deposition for the following figures:

Figure 4BDF, 5BEG, 6F, 7CDEFGHI, 8A-M, S2G, S3DEG, S4BE, S5E-J, S6GH, S7AC, S8A-M

f) Please cite the location of the data clearly in all relevant main and supplementary Figure legends, e.g. “The data underlying this Figure can be found in S1 Data” or “The data underlying this Figure can be found in https://doi.org/10.5281/zenodo.XXXXX”

g) We require the original, uncropped and minimally adjusted images supporting all blot and gel results reported in the Figures 3AB, 6ABCDEGHI, 7J, S3A, S5CD, S6A-FIJK

We will require these files before a manuscript can be accepted so please prepare and upload them now. Please carefully read our guidelines for how to prepare and upload this data: https://journals.plos.org/plosbiology/s/figures#loc-blot-and-gel-reporting-requirements

h) Please ensure that your Data Statement in the submission system accurately describes where your data can be found and is in final format, as it will be published as written there. Please note that by journal policies, all data should be provided either as Supplementary Material or on a repository such as Zenodo. Anything that is not in the Supplementary Material should be deposited in a stable, community-accepted repositories, or somewhere like zenodo, and the accession numbers provided. For microscopy, the "image data resource" si a good option, I think https://idr.openmicroscopy.org/about/index.html - information on how this can be done can be found here https://idr.openmicroscopy.org/about/submission.html.

i) Per journal policy, if you have generated any custom code during the course of this investigation, please make it available without restrictions upon publication. Please ensure that the code is sufficiently well documented and reusable, and that your Data Statement in the Editorial Manager submission system accurately describes where your code can be found.

We expect to receive your revised manuscript within two weeks. 

*Published Peer Review History*

*Press*

Sincerely,

Melissa

Melissa Vazquez Hernandez, Ph.D.

Associate Editor

PLOS Biology

REVIEWERS' COMMENTS

Reviewer #1: I have no further question on this manuscript.

Reviewer #3: The authors have thoroughly addressed all my concerns.

---

## [Editor Report · Decision Letter 3]

3 Oct 2024

Dear Dr Rampalli,

Thank you for the submission of your revised Research Article "A cytoplasmic form of EHMT1N methylates viral proteins to enable inclusion body maturation and efficient viral replication" for publication in PLOS Biology. On behalf of my colleagues and the Academic Editor, Frank Kirchhoff, I am pleased to say that we can in principle accept your manuscript for publication, provided you address any remaining formatting and reporting issues. These will be detailed in an email you should receive within 2-3 business days from our colleagues in the journal operations team; no action is required from you until then. Please note that we will not be able to formally accept your manuscript and schedule it for publication until you have completed any requested changes.

IMPORTANT: We need a clarification for the Data Availability Statement. You said "Data from this study is available at CSIR-IGIB and authors may be contacted at shravanti@igib.res.in, shravanti@igib.in" We do not allow this unless is only raw data. If what you mean that the raw data is available with you but the relevant material is available in the Supplementary and at BioImage Archive, then please change your statement to "The summary data underlying the figures are in the SI, the raw imaging data at XXX in the Bioimage archive, and the rest of the raw data is available at CSIR-IGIB and authors may be contacted at shravanti@igib.res.in, shravanti@igib.in". I was also not able to find the material at BioImage Archive. Please make it available to move forward with the publication.

I have asked my colleagues to include this request alongside their own.

PRESS

Sincerely, 

Melissa

Melissa Vazquez Hernandez, Ph.D., Ph.D.

Associate Editor

PLOS Biology
